# Dual-IPO: Dual-Iterative Preference Optimization for Text-to-Video Generation

**Xiaomeng Yang**[2†] **Mengping Yang**[1,2†] **Jia Gong**[2] **Luozheng Qin**[2] **Zhiyu Tan**[1,2*] **Hao Li**[1,2*]

[1]Fudan University, [2]Shanghai Academy of AI for Science

## Abstract

Recent advances in video generation have enabled thrilling experiences in producing realistic videos driven by scalable diffusion transformers. However, they usually fail to produce satisfactory outputs that are aligned to users' authentic demands and preferences. In this work, we introduce Dual-Iterative Optimization (Dual-IPO), an iterative paradigm that sequentially optimizes both the reward model and the video generation model for improved synthesis quality and human preference alignment. For the reward model, our framework ensures reliable and robust reward signals via CoT-guided reasoning, voting-based self-consistency, and preference certainty estimation. Given this, we optimize video foundation models with guidance of signals from reward model's feedback, thus improving the synthesis quality in subject consistency, motion smoothness and aesthetic quality, etc. The reward model and video generation model complement each other and are progressively improved in the multi-round iteration, without requiring tediously manual preference annotations. Comprehensive experiments demonstrate that the proposed Dual-IPO can effectively and consistently improve the video generation quality of base model with various architectures and sizes, even help a model with only 2B parameters surpass a 5B one. Moreover, our analysis experiments and ablation studies identify the rational of our systematic design and the efficacy of each component.

## 1 Introduction

Video generation Brooks et al. (2024); Kuaishou (2024); hailuo (2024); Kong et al. (2024); Yang et al. (2024b); Tan et al. (2025) has experienced tremendous advancement in recent years. Their technical breakthroughs, driven by scalable model architectures (*e.g., diffusion transformer Peebles & Xie (2023)*), web-scale data and massive computation, have revolutionized traditional concent creation in various domains like movie production Zhang et al. (2025), story Zhou et al. (2024) and scene generation Guo et al. (2025), *etc.* Despite their success, existing video generation models still fall short in producing user-satisfied outputs with consistent subject, smooth motion and compelling aesthetic. This identifies a considerable gap between the quality of current models and the authentic demands of users in practice.

To improve synthesis quality and alignment of human preference, post-training techniques like Direct Preference Optimization (DPO) Rafailov et al. (2023); Liu et al. (2025b) and Kahneman-Tversky Optimization (KTO) Ethayarajh et al. (2024) are employed. However, preference learning typically requires training on large annotated datasets of human preferences, which are laborious to construct. Alternatively, external reward models, such as HPS Wu et al. (2023), ImageReward Xu et al. (2023), VideoAlign Liu et al. (2025a) and VisionReward Xu et al. (2024), can be utilized to evaluate the quality of images and videos for preference annotations. Nonetheless, these reward models often exhibit a distribution mismatch Kim et al. (2024) across different video generation models and evaluation criteria, making the reward signals unreliable for human preference alignment (see Tab. 1). Additionally, aligning video generation models on fixed offline preference datasets could easily cause overfitting and even collapsed to the preference signals Pal et al. (2024);

---

[†]Equal Contribution, [*]Corresponding Authors.
Our source code is available at `https://github.com/SAIS-FUXI/IPO`.

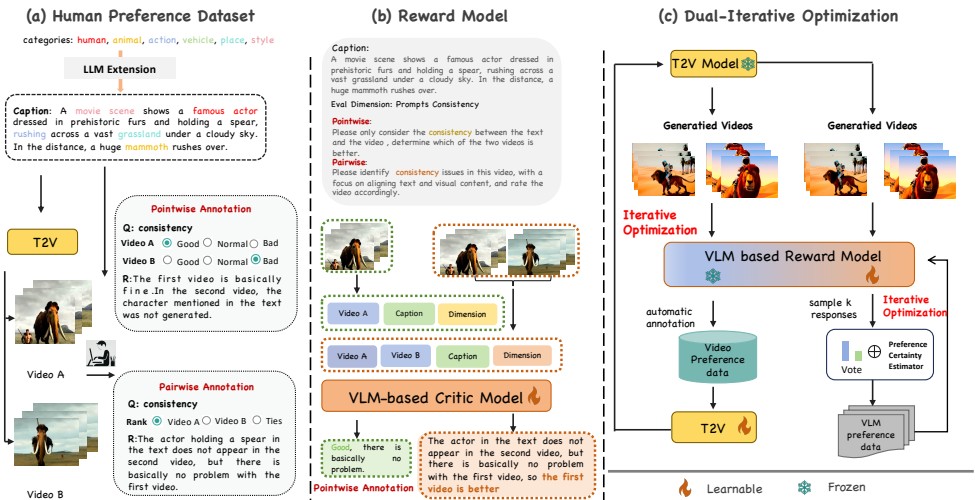

Figure 1: Overview of our proposed dual-iterative preference optimization framework. (a) Human preference dataset used to initially train our reward model. (b) Reward model that learns to automatically annotate generated videos with preference labels. (c) Dual-iterative optimization, which incorporates the reward model to iteratively optimize video foundation model, thus improving the synthesis quality and human alignment.

Feng et al. (2024), degrading the synthesis quality and model generalization Park et al. (2024); Chen et al. (2024b); Azar et al. (2024). Furthermore, the reward signal should evolve with the model since the artifacts are becoming subtler throughout the training, otherwise the reward might be biased and lead to instable optimization Dubey et al. (2024).

To address the aforementioned challenges, this paper proposes *Dual-iterative preference optimization (Dual-IPO)*, a post-training framework for improved synthesis quality and human alignment via iterative optimization on both the reward model and the video generation model. For the iterative refinement of our reward model, we propose *self-refined preference optimization* to unlock stronger reasoning capabilities from Vision-Language Models (VLMs) with a small set of Chain-of-Thought (CoT) guided annotated data, enabling high-quality and efficient reward modeling with minimal supervision. Then, we design a voting-based *self-consistency mechanism* across multiple inference paths to improve the accuracy and scalability of labeled rewards. Furthermore, a *preference certainty estimator* is developed to filter low-confidence results, ensuring reliable pseudo-labels for further training of the reward model. Such iterative self-labeling cycle strengthens the reward model's generalization ability and robustness over time, ensuring reliable rewards for training video generation models. Regarding the iterative optimization of video generation models, the latest model is used to generate new videos in each iteration. These generated videos are scored by the current reward model, serving as updated reward signals for the next optimization iteration. Notably, our reward model supports both pairwise and pointwise preference annotation, facilitating flexible deployment under different optimization settings, *i.e.*, pairwise for Diffusion-DPO and pointwise for Diffusion-KTO. The two component complement each other and together enhance the overall quality and user preference alignment of generated outputs. Besides, only a small amount of annotated preferences are required as cold start data for training our reward model, suggesting satisfactory data efficiency.

To testify the effectiveness of our proposed framework, we conduct extensive experiments on video generation models with different architectures (*i.e.,* cross-attention DiT based Wanx Wan et al. (2025) and MMDiT based CogVideo Yang et al. (2024b)), different model scales (*i.e.,* 2B and 5B). Experimental results demonstrate that our method reflects substantial performance gains across different baseline models in various aspects, including subject consistency, motion smoothness, visual aesthetic, *etc*. Moreover, the synthesis performance could be consistently improved with our proposed multi-round iterative optimization, indicating that both the reward model and the video generation model are progressively refined through each iteration. To sum up, our primary contributions are summarized blow:

- **A dual-iterative preference optimization framework for improving video generation models.** By updating the reward model and the video generation model in a multi-round manner, we successfully improve the synthesis quality and human preference alignment.

- **A self-refined preference optimization strategy for reliable reward models.** With CoT-guided reasoning, voting-based self-consistency, and preference certainty estimation, we ensure high-confidence and accurate reward signals iteratively, thus improving the generalization and robustness of the reward model. Notably, both pairwise and pointwise rewards are supported, enabling flexible deployment for different alignment strategies.

- **A multi-round scheme for consistently improvement on video quality.** Leveraging these reliable rewards, we refine T2V models iteratively across multiple aspects (semantic, motion, aesthetic), consistently leading to better alignment and stronger generalization.

- **State-of-the-art performance under various settings.** Extensive experiments show that our framework significantly advances several baseline models with different sizes. In particular, we empower a 2B CogVideoX model to outperform a $2.5\times$ larger 5B variant, demonstrating superior alignment efficiency and effectiveness.

## 2 RELATED WORK

**Text-to-video generation.** Recent advances in video generation have been largely driven by diffusion models Ho et al. (2020); Sohl-Dickstein et al. (2015); Song et al. (2020); Zhang et al. (2022), achieving remarkable improvements in diversity, fidelity, and quality Chen et al. (2023); Esser et al. (2023); Ho et al. (2022b); Liu et al. (2023a); Ruan et al. (2023); Wang et al. (2025); Xing et al. (2023). Scaling model size and training data further enhance performance Ho et al. (2022a); Singer et al. (2022); Wu et al. (2021). In text-to-video (T2V) generation, pretraining from scratch requires substantial computational resources and large-scale datasets Gen-3 (2024); Luma (2024); Kuaishou (2024); Brooks et al. (2024); Kong et al. (2024). Current methods Blattmann et al. (2023); Wang et al. (2023); Guo et al. (2023) extend text-to-image models by incorporating spatial and temporal attention modules, often using joint image-video training Ma et al. (2024), with Guo et al. (2023) proposing a plug-and-play temporal module for personalized image models. Remarkable visual quality and temporal consistency have been achieved by Zhang et al. (2023); Chen et al. (2024a), and diffusion transformers (DiT) Brooks et al. (2024); Yang et al. (2024b); Chen et al. (2024a) enhance spatiotemporal coherence and scalability, identifying the great potential of transformer architectures. Despite their success, they often fail to produce user-preferred videos in practice.

**Post-training for human preference alignment.** Post-training is widely adopted in Large Language Models (LLMs)Ouyang et al. (2022); Ramamurthy et al. (2022); Zheng et al. (2023). RLHFOuyang et al. (2022) aligns models via reward functions learned from comparisons. RRHF Yuan et al. (2023b) extends supervised fine-tuning, and SLiC Zhao et al. (2023) uses offline RL data. DPO Rafailov et al. (2023) improves stability by directly optimizing preference data, while KTO Ethayarajh et al. (2024) adopts a prospect-theoretic view. RSO Liu et al. (2023b) leverages rejection sampling for better alignment. Post-training has also been applied to image foundation models Clark et al. (2023); Prabhudesai et al. (2024a); Xu et al. (2023). For diffusion models, Diffusion-DPO Wallace et al. (2023) extends DPO to optimize likelihood, while Diffusion-KTO Li et al. (2025) maximizes expected utility without pairwise preferences. However, post-training for video generation remains underexplored. Existing efforts focus on reward design. InstructVideo Yuan et al. (2023a) proposes temporal decaying rewards (TAR) to prioritize central frames. T2V-Turbo Li et al. (2024) improves ODE solvers using motion priors. VADER Prabhudesai et al. (2024b) fine-tunes with vision-based reward gradients, boosting generalization. Despite these, gains remain limited, and the full potential of post-training is yet to be realized. In this paper, we propose a dual-optimization strategy to iteratively update the reward model and the video generation model. They complement each other and together deliver improved output quality and preference alignment.

## 3 METHOD

### 3.1 METHOD OVERVIEW

Fig. 1 presents the overall framework of our method. First, we collect a small amount of annotated user preferences for initially training our reward model. In this phase, both pairwise and pointwise annotations are constructed, enabling flexible alignment strategies like DPO with pairwise data and KTO with pointwise data. After training on the human preference dataset, we incorporate the reward model and the T2V generation model in the iterative optimization loop, performing dual-optimization to fully unlock their potentials. We detail each component of our method below.

### 3.2 SELF-REFINED PREFERENCE OPTIMIZATION

We propose *self-refined preference optimization (SRPO)*, a self-iterative training loop that progressively enhances the reward model with minimal supervision.

**CoT-guided annotation.** We start by constructing a small set of Chain-of-Thought (CoT) guided annotations, unlocking the potential of a Vision-Language Model (VLM) for structured reasoning and reliable preference construction. Specifically, we employ CogVideoX-2B model Yang et al. (2024b) to generate videos from a diverse prompts built by Qwen2.5 Yang et al. (2024a). These prompts span over ten semantic categories (e.g., Human, Vehicle, Architecture) and multiple motion types (details are in the appendix). Each prompt yields four videos from different seeds, and the dataset is iteratively expanded using an optimized regeneration strategy. We collect both pairwise ranked and pointwise scored preference data along three dimensions: text-video consistency, content faithfulness, and motion smoothness following Tan et al. (2024b).

**Self-consistency via multi-path inference.** For preference queries with deterministic answers, voting across multiple reasoning paths has been shown to improve reliability Prasad et al. (2024); Wang et al. (2022). Following this, SRPO performs multi-path inference with CoT-enabled reward model to scale the preference data. Concretely, we adopt a self-consistency mechanism that aggregates responses by counting answer frequencies, automatically constructing stable pseudo preference labels and reducing noise from individual paths.

**Preference Certainty Estimator(PCE).** While self-consistency improves pseudo-label quality, noise can still arise due to randomness in VLM outputs and lack of reasoning trace inspection. To further filter unreliable labels, we introduce a preference certainty estimator (PCE) based on the reward formulation used in DPO Rafailov et al. (2023). Given a prompt $x$, let $y_w$ and $y_l$ denote the preferred and less-preferred videos in a pair. Formally, DPO models the preference probability Zhu et al. (2025); Kim et al. (2024) as:

$$\mathbb{P}_\theta(y_w \succ y_l \,|\, x) = \sigma\left(\beta \log \frac{\pi_\theta(y_w \mid x)}{\pi_{\text{ref}}(y_w \mid x)} - \beta \log \frac{\pi_\theta(y_l \mid x)}{\pi_{\text{ref}}(y_l \mid x)}\right), \tag{1}$$

where $\pi_\theta$ and $\pi_{\text{ref}}$ are the trained and reference models, respectively, and $\beta$ is a temperature parameter. After DPO learning, the model implicitly learns a reward-like scoring to indicate the model's preference $y$ over the baseline:

$$R(y \mid x) = \log \frac{\pi_\theta(y \mid x)}{\pi_{\text{ref}}(y \mid x)}. \tag{2}$$

We define the average reward across all samples in the current dataset as $Q = \mathbb{E}_{(x,y)}[R(y \mid x)]$. To estimate the confidence that $y_w$ is truly better than the average, we define our certainty metric as:

$$\text{PCE}(y_w \mid x) = \mathbb{P}_\theta\left(R(y_w \mid x) > Q\right). \tag{3}$$

Intuitively, this measures how likely the chosen sample is better than a typical sample in the current distribution. We filter out weak or uncertain preferences by only retaining data points where $\text{PCE}(y_w \mid x) > 0.5$. This criterion ensures that the selected preference aligns not just with relative ranking, but also exceeds the global quality threshold, improving label robustness under distribution shift.

**Iterative Pseudo-Label Refinement.** The refined pseudo-labels are used to train the critic model, which is then applied to newly generated video samples for the next iteration. This bootstrapping process allows the critic to progressively improve its alignment with human preferences and generalize to evolving data distributions.

We adopt the DPO objective Rafailov et al. (2023), which increases the preference margin between the preferred sample $y_w$ and the less-preferred $y_l$ given the same input $x$ (e.g., text prompt). The training loss is defined as:

$$\mathcal{L}_{\text{DPO}}(\theta) = -\mathbb{E}_{x,y_w,y_l} \left[ \log \sigma \left( \beta \log \frac{p_\theta(y_w \mid x)}{p_{\text{ref}}(y_w \mid x)} - \beta \log \frac{p_\theta(y_l \mid x)}{p_{\text{ref}}(y_l \mid x)} \right) \right]$$

where $\sigma(\cdot)$ is the sigmoid function, $\beta$ is a temperature parameter, and $p_{\text{ref}}$ is a frozen reference model. To enhance robustness, we further introduce the preference certainty estimator (PCE), defined as:

$$\text{PCE}(y_w \mid x) = \mathbb{P}_\theta \left( R(y_w \mid x) > Q \right), \tag{4}$$

where $R(y \mid x) = \log \frac{p_\theta(y|x)}{p_{\text{ref}}(y|x)}$ is the learned reward score, and $Q = \mathbb{E}_{x,y}[R(y \mid x)]$ is the mean reward over the dataset. Consequently, the final loss incorporates PCE as a confidence weight:

$$\mathcal{L}_{\text{SRPO}}(\theta) = \mathbb{E}_{x,y_w,y_l} \left[ \text{PCE}(y_w \mid x) \cdot \mathcal{L}_{\text{DPO}}(\theta) \right]. \tag{5}$$

This formulation emphasizes reliable preferences during training and suppresses noisy supervision, leading to a more stable and generalizable reward model.

## 3.3 ITERATIVE ALIGNMENT FOR VIDEO GENERATION

After building an reliable reward model, we turn to align text-to-video (T2V) models with its feedback. Unlike one-time alignment methods that rely on offline fixed preference datasets, we iteratively leverage dynamic feedback from the self-improving reward model to provide robust and adaptive preference supervision throughout training.

**Optimization loop.** Our core design is an iterative training loop between the T2V model and the reward model. In each iteration, the generator produces a batch of videos from diverse prompts. The reward model then evaluates these videos and produce pairwise data for DPO optimization and pointwise data for KTO optimization. To ensure training stability and efficient convergence, we monitor key indicators including the VBench metrics and training loss curves throughout the training process. When performance degradation (e.g., drop in semantic fidelity or motion quality) or abnormal loss dynamics are detected, the framework automatically triggers a critic update via the self-refined preference optimization procedure. This ensures more accurate and distribution-aware reward model, which in turn to re-score generated videos and reconstruct updated preference datasets for the next iteration. In this way, we can effectively mitigate reward misalignment and overfitting, enabling progressive preference-driven refinement of the T2V model.

**Different preference strategies.** We support both pairwise and pointwise alignment paradigms. For pairwise supervision, we adopt Diffusion-DPO Wallace et al. (2023), which extends Direct Preference Optimization to the diffusion setting. Given preference pairs $(x_0^w, x_0^l)$ scored by the critic, Diffusion-DPO minimizes the following loss:

$$L_{\text{dpo}}(\theta) = -\mathbb{E}_{(x_0^w,x_0^l)\sim D,\, t\sim \mathcal{U}(0,T)} \log \sigma \Big( -\beta T \omega(\lambda_t) \cdot \big[ \tag{6}$$
$$\|\epsilon^w - \epsilon_\theta(x_t^w,t)\|^2 - \|\epsilon^w - \epsilon_{\text{ref}}(x_t^w,t)\|^2 - (\|\epsilon^l - \epsilon_\theta(x_t^l,t)\|^2 - \|\epsilon^l - \epsilon_{\text{ref}}(x_t^l,t)\|^2) \big] \Big).$$

where $\omega(\lambda_t)$ is a time-dependent weighting function, and $\epsilon^w$, $\epsilon^l$ are noise vectors from the forward process. To improve generalization, we add two auxiliary negative log-likelihood (NLL) terms:

$$\mathcal{L}_{\text{total}}^{\text{DPO}}(\theta) = L_{\text{dpo}}(\theta) + \lambda_1 \cdot \mathbb{E}_{\mathcal{D}^{\text{sample}}} \big[ -\log p_\theta(y^w|x) \big] + \lambda_2 \cdot \mathbb{E}_{\mathcal{D}^{\text{real}}} \big[ -\log p_\theta(y|x) \big], \tag{7}$$

where $\mathcal{D}^{\text{sample}}$ contains preference samples, and $\mathcal{D}^{\text{real}}$ contains real videos for regularization.

For pointwise alignment, we incorporate Diffusion-KTO Li et al. (2025), which optimizes a nonlinear utility-based objective without requiring preference pairs. The core loss is defined as:

$$L_{\text{kto}}(\theta) = \mathbb{E}_{x_0\sim\mathcal{D},\, t\sim\text{Uniform}(0,T)} \left[ U \left( w(x_0) \left( \beta \log \frac{\pi_\theta(x_{t-1}|x_t)}{\pi_{\text{ref}}(x_{t-1}|x_t)} - Q_{\text{ref}} \right) \right) \right], \tag{8}$$

where $w(x_0) \in \{\pm 1\}$ denotes desirability, and the reference value $Q_{\text{ref}}$ is:

$$Q_{\text{ref}} = \beta \cdot \mathbb{D}_{\text{KL}} \left[ \pi_\theta(a|s) \, || \, \pi_{\text{ref}}(a|s) \right]. \tag{9}$$

To stabilize training, we further regularize the loss using real-world data as:

$$\mathcal{L}_{\text{total}}^{\text{KTO}}(\theta) = L_{\text{kto}}(\theta) + \lambda \cdot \mathbb{E}_{\mathcal{D}^{\text{real}}} \left[ -\log p_\theta(y|x) \right]. \tag{10}$$

Note that these two optimization paths are activated independently within our optimization process according to the available preference data format, allowing flexible deployment under varied alignment strategies. Our dual-iterative optimization framework offers several key advantages: 1) we dynamically align the reward model drift and generator distributional shifts through dual optimization, improving training robustness and stability; 2) we significantly reduce requirements on large-scale human annotations by leveraging critic self-refinement; 3) we provide a unified yet flexible framework for handling both pointwise and pairwise preference data; and 4) we achieve stronger generalization and stability by continuously aligning generation behavior with up-to-date human preferences.

## 4 EXPERIMENTS

### 4.1 IMPLEMENTATION DETAILS

We fine-tune pre-trained VILA Lin et al. (2023) (13B/40B) models as reward models, leveraging their strong video-text understanding and instruction-following abilities. A small set of Chain-of-Thought (CoT) annotations is used for warm-start, followed by 5 epochs of training (LR 2e-5, batch size 4 on 32 GPUs). Videos are represented by 16 uniformly sampled frames with padding-based preprocessing. Other settings follow VILA defaults. We further apply Self-Refined Preference Optimization (SRPO) with CoT-guided inference, self-consistency voting, and confidence filtering, generating 20K pseudo-labels per iteration for iterative fine-tuning.

**T2V alignment training.** For generator alignment, the critic assesses semantic fidelity and temporal consistency under the Dual-IPO framework with Diffusion-DPO and Diffusion-KTO. Training uses a batch size of 512 and LR 2e-6 (raised to 2e-5 after stabilization) across 128 GPUs. For each caption, the T2V model generates 8 candidates, with the top-1 and bottom-1 (by critic ranking) forming preference pairs. Each iteration constructs about 100K such pairs, and a single training round takes on average two weeks.

**Datasets.** To construct large-scale and diverse training data, we first build a pool of structured elements (subjects, attributes, spatial relations, actions), which are randomly composed and expanded into natural language captions using an LLM, producing about one million captions. The target T2V model then generates corresponding videos for iterative optimization. In addition, a subset of real video-text pairs from VidGen-1M Tan et al. (2024a) is incorporated for joint fine-tuning to enhance diversity and grounding in natural distributions.

**Evaluation.** For evaluation, we use both automatic metrics and human assessments. We report the overall and per-dimension scores from the VBench benchmark Huang et al. (2024), including semantic consistency, motion smoothness, and visual quality. To complement these metrics, we conduct user studies to verify synthesis quality.

### 4.2 EXPERIMENTAL RESULTS

**A customized reward model is necessary for reliable alignment.** We first evaluate the effectiveness and generalizability of our reward model by measuring alignment with human preference labels. We randomly sample 1,000 videos from CogVideoX-2B and WanX-1.3B, and manually annotate them as a reference set. Our critic is compared against three open-source models: VideoScore He et al. (2024),

Table 1: Comparison of reward models by human preference accuracy and downstream VBench score after DPO training.

| Reward Models | Accuracy (%) | VBench |
|---|---|---|
| VideoScore | 63.58 | 80.87 |
| VideoAlign | 65.21 | 81.27 |
| VideoReward | 68.44 | 81.31 |
| Ours | **81.33** | **81.54** |

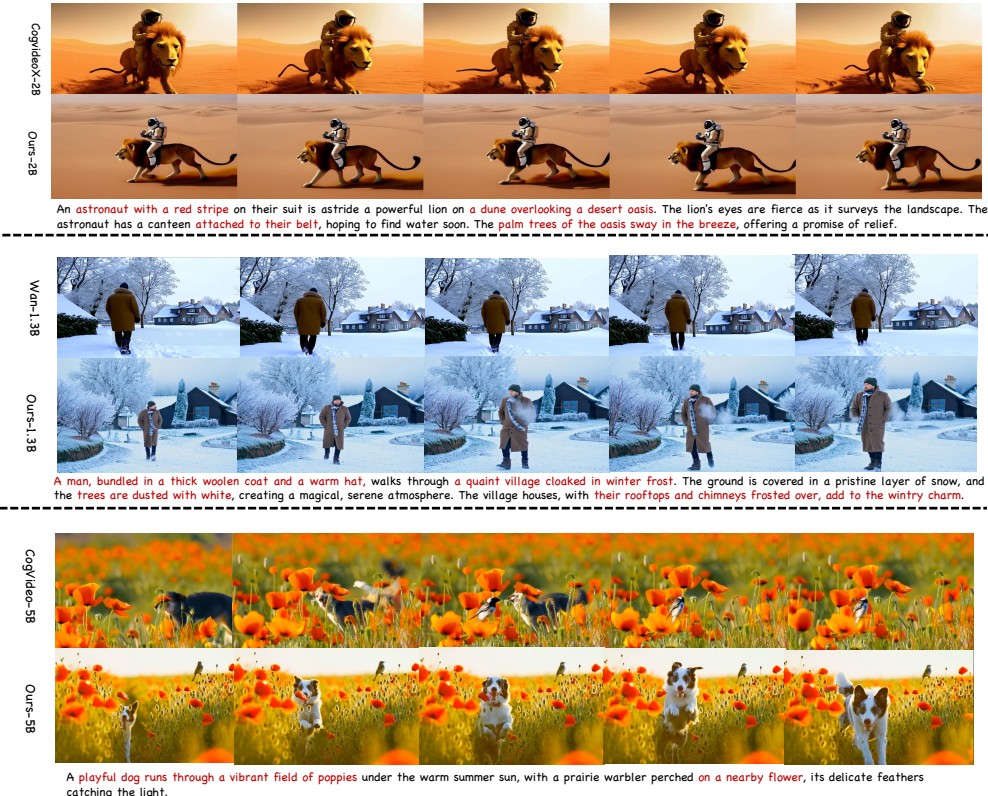

Figure 2: Qualitative Comparison of our proposed Dual-IPO model and the baseline CogVideoX-2B model. We can see Dual-IPO improves the baseline in the aspects of prompt following (the 1st line), aesthetic quality (the 2nd line), subject consistency (the 3rd line) and motion smoothness (the 4th line). This demonstrates the efffectiveness of our Dual-IPO for enhancing the generation capability of video found models. Additional visualization results are available in the supplementary materials.

VideoAlign Liu et al. (2025a), and VisionReward Xu et al. (2024). As shown in Tab. 1, our reward model achieves **81.33%** accuracy, significantly outperforming VisionReward (**68.44%**), VideoAlign (**65.21%**), and VideoScore (**63.58%**). The weaker generalization of these models is due to their training on different data and annotation schemes.

We further use each critic to construct preference datasets for fine-tuning CogVideoX-2B via Diffusion-DPO. As shown in Tab. 1, our critic boosts VBench from **80.91** to **81.54** (**+0.67**), while VideoScore slightly degrades performance, and VideoAlign and VisionReward yield only minor gains around **+0.3**. This shows critic models must be tailored to the specific T2V generator they supervise. Notably, our reward model is trained on just **6,000** annotated pairs, fewer than other models, yet achieves strong alignment. We attribute this to Chain-of-Thought reasoning and VLM-based inference, which offer richer supervision from limited data.

**Comparison with current sota.**    We evaluate our method on VBench (Tab.3), where our optimized **2B** model surpasses the baseline **5B** across all dimensions, showing that Dual-IPO post-training effectively boosts generation quality in smaller models. As shown in Tab.2, Dual-IPO consistently improves motion, aesthetics, and consistency during iterative training, demonstrating both stability and effectiveness. These gains also reflect better prompt alignment and fewer motion artifacts in generated videos.

To further validate our generalization, we apply our model to both CogVideoX-5B and Wan-1.3B. As reported in Tab. 3, after three iterations, CogVideoX-5B improves from 81.61 to 84.63 on VBench, and Wan-1.3B increases from 84.26 to 86.28. These gains are evident in both visual quality and prompt consistency. We also evaluate the final models after five Dual-IPO in Tab. 5, it turns out that

Table 2: Ablation studies on the iteration number adopted in Dual-IPO. The evaluation is performed on VBench.

| Models | Total Score. | Quality Score. | Semantic Score. | Subject Consist. | Background Consist. | Temporal Flicker. | Motion Smooth. | Aesthetic Quality |
|---|---|---|---|---|---|---|---|---|
| CogVideoX-2B | 80.91 | 82.18 | 75.83 | 96.78 | 96.63 | 98.89 | 97.73 | 60.82 |
| CogVideoX-2B-IPO$_1$ | 81.69 | 82.87 | 77.01 | 96.77 | 97.01 | 99.01 | 97.84 | 61.55 |
| CogVideoX-2B-IPO$_2$ | 82.42 | 83.53 | 77.97 | 96.81 | 97.23 | 99.21 | 98.03 | 62.35 |
| CogVideoX-2B-IPO$_3$ | 82.74 | 83.92 | 78.00 | 96.79 | 97.48 | 99.35 | 98.17 | 62.31 |

Table 3: Dual-IPO under various baselines.

| Model | Total | Quality | Semantic |
|---|---|---|---|
| CogVideoX-2B | 80.91 | 82.18 | 75.83 |
| CogVideoX-2B-IPO-3 | **82.74** | **83.92** | **78.00** |
| CogVideoX-5B | 81.61 | 82.75 | 77.04 |
| CogVideoX-5B-IPO-3 | **84.63** | **85.40** | **81.54** |
| Wan-1.3B | 84.26 | 85.30 | 80.09 |
| Wan-1.3B-IPO-3 | **86.28** | **86.38** | **85.87** |

Table 4: Ablation studies: DPO vs KTO.

| Method | Total | Quality | Semantic |
|---|---|---|---|
| CogVideoX-2B | 80.91 | 82.18 | 75.83 |
| CogVideoX-2B-KTO$_3$ | 82.47 | 83.56 | 78.08 |
| CogVideoX-2B-DPO$_3$ | 82.42 | 83.53 | 77.97 |

Table 5: Comparison with sota T2V models on VBench.

| Model | Total | Quality | Semantic |
|---|---|---|---|
| Vidu Q1 | 87.41 | 87.28 | 87.94 |
| Open-Sora-2.0 | 84.34 | 85.40 | 80.12 |
| Sora | 84.28 | 85.51 | 79.35 |
| CausVid | 84.27 | 85.65 | 78.75 |
| Wan2.1 | 86.22 | 86.67 | 84.44 |
| CausVid | 83.88 | 85.21 | 78.57 |
| HunyuanVideo | 83.24 | 85.09 | 75.82 |
| CogVideoX-5B | 82.01 | 82.72 | 79.17 |
| +Ours | **86.57** | **87.00** | **84.84** |
| CogVideoX-2B | 80.91 | 82.18 | 75.83 |
| +Ours | **82.74** | **83.92** | **78.00** |
| Wan-1.3B | 84.26 | 85.30 | 80.09 |
| +Ours | **88.32** | **87.83** | **90.25** |

our model yields consistent performance gains, demonstrating both the effectiveness and generalizability of our proposed framework across architectures and scales.

**Qualitative results.** To further assess our approach, we present qualitative comparisons in Fig. 2. The optimized model consistently produces videos with better semantic fidelity, accurately reflecting prompt intent and context. It also improves temporal consistency by reducing artifacts and abrupt transitions. Additionally, the outputs show enhanced realism with fewer structural errors, demonstrating the effectiveness of Dual-IPO.

**Effectiveness of Self-Refined Preference Optimization (SRPO).** To evaluate SRPO's role in improving critic generalization, we conduct controlled ablations during the training of CogVideoX-2B in Tab. 6. Specifically, we examine model performance between the third and fourth IPO iterations under two conditions: (1) keeping the critic model fixed, and (2) updating the critic via SRPO. We can observe that without critic update, performance declines

Table 6: Ablation study on SRPO during the third to fourth Dual-IPO iteration on the CogVideoX-2B baseline.

| Setting | Critic Update | SRPO loss | PCE | VBench Score |
|---|---|---|---|---|
| Baseline | ✗ | — | — | 82.74 |
| No update | ✗ | — | — | 82.33 |
| SRPO (full) | ✓ | ✓ | ✓ | **82.91** |
| SRPO loss | ✓ | ✗ | ✓ | 82.83 |
| PCE | ✓ | ✓ | ✗ | 82.69 |

from **82.74** to **82.33**, indicating overfitting or misalignment. In contrast, applying SRPO increase the performance from **82.74** to **82.91**, demonstrating the necessity and efficacy of our proposed reward model in-the-loop optimization.

We ablate SRPO components. Removing PCE with SRPO loss causes minimal change (82.69), suggesting limited impact. Replacing SRPO loss with DPO, even with PCE, drops performance to **82.83**, confirming SRPO loss is crucial for critic robustness and signal quality. These results underscore the importance of reward modeling, critic refinement, and self-refinement for stable alignment.

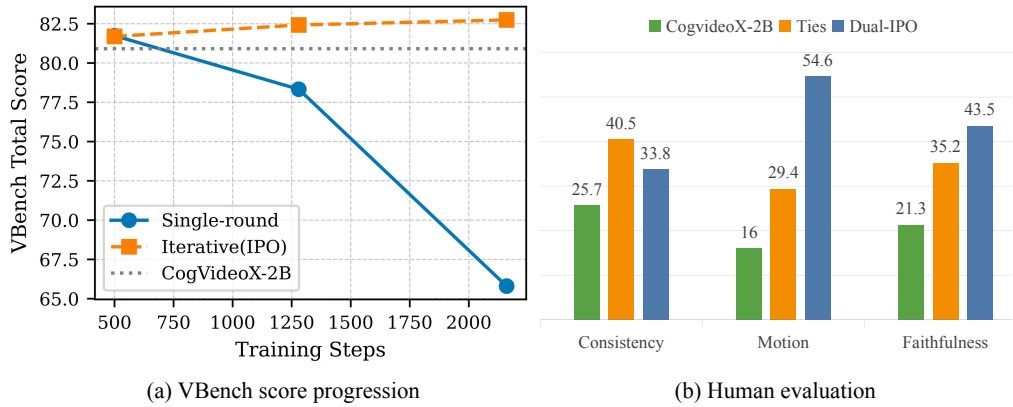

(a) VBench score progression        (b) Human evaluation

Figure 3: Comparison of IPO and baselines: (a) VBench shows DPO degrades, while Dual-IPO improves steadily. (b) Human evaluation compares Dual-IPO, CogVideoX-2B, and Ties on consistency, motion, and faithfulness.

## 4.3 EFFECTIVENESS OF ITERATIVE ALIGNMENT

**Why multi-round iteration matters.** To evaluate iterative optimization, we compare Dual-IPO with a static baseline using Diffusion-DPO without updating the preference data or the reward model. As shown in Fig.3a, the static baseline improves briefly but degrades due to overfitting and misaligned rewards, while Dual-IPO maintains steady performance gains with progressive data improvement and reward optimization. Tab.3 and Tab. 2 confirm that Dual-IPO outperforms the static method in overall VBench scores and all quality dimensions, with consistent improvements in motion, aesthetics, and semantic consistency.

**Benefits of incorporating real video data.** We examine whether real-world video data improves post-training. Using VidGen-1M Tan et al. (2024a) with diffusion loss and preference optimization, we observe consistent gains in total, quality, and semantic scores (Tab. 7). In contrast, real data alone slightly hurts performance, suggesting it is most effective when combined with preference guidance. We leave further analysis to future work.

Table 7: Ablation study on the effects of incorporating real video data during Dual-IPO fine-tuning.

| Model | Total | Quality | Semantic |
|---|---|---|---|
| CogVideoX-2B | 80.91 | 82.18 | 75.83 |
| only real data sft | 80.63 | 81.79 | 76.02 |
| w/o real data sft | 81.17 | 82.33 | 76.53 |
| w/ real data sft | 81.69 | 82.87 | 77.01 |

**Comparing DPO and KTO within Dual-IPO.** To assess flexibility, we compare DPO and KTO within Dual-IPO (Tab. 4). Both outperform the baseline. KTO slightly improves Total and Semantic scores, while DPO leads in Quality, showing Dual-IPO supports diverse alignment. KTO's more stable semantic gains suggest better high-level consistency, confirming the adaptability of Dual-IPO and effectiveness of KTO for fine-grained optimization.

**Human evaluation.** Fig. 3b shows the results of human visual evaluation that compare our Dual-IPO with the CogVideo-2B baseline. Obvisouly, Dual-IPO outperforms the baseline across all dimensions. In "Consistency," both models perform equally in 40.5% of cases, suggesting the baseline handles text alignment reasonably well. However, Dual-IPO shows clear gains in "Motion" and "Faithfulness," generating more natural movement and better text alignment. Full evaluation metrics are in the appendix.

## 5 CONCLUSION

In this paper, we present a novel Iterative Preference Optimization framework for improvement video foundation models from the pespective of post training. In particular, Dual-IPO collects a

human preference dataset and trains a critic model for automatically labeling the generated videos from base model. Given this, Dual-IPO efficiently optimizes the base model in an iterative manner. By this way, Dual-IPO can effectively align the base model with human preference, leading to enhanced video results in motion smoothness, subject consistency and aesthetic quality, etc. In addition, we adapt Dual-IPO with different optimization strategies: DPO and KTO, both achieve impressive improvements. This further verifies the flexibility of Dual-IPO. Thorough experiments on VBench demonstrates the efficacy of Dual-IPO.

## ACKNOWLEDGMENTS

This work was supported by AI for Science Program, Shanghai Municipal Commission of Economy and Informatization (Grant No. 2025-GZL-RGZN-BTBX-02017).

## ETHICS STATEMENT

This work focuses on developing a dual-iterative optimization framework for text-to-video generation. All datasets used are either publicly available or synthetically generated through large-scale text-to-video models. Only a small set of human-labeled preference data is employed for initial training, after which the framework relies mainly on automated pseudo-labeling. No sensitive, private, or personally identifiable information is used, and no human subjects are involved in ways that raise ethical concerns. The proposed method aims solely to improve video generation quality and alignment with human preferences. Therefore, we believe that this research does not pose ethical risks.

## REPRODUCIBILITY STATEMENT

We provide detailed descriptions of our framework, including algorithmic formulations, optimization objectives, and training procedures. Implementation details such as learning rates, batch sizes, GPU usage, model architectures, and dataset construction are explicitly reported. Extensive experiments are conducted with multiple baselines, ablation studies, and both automatic and human evaluations. All benchmarks employed (e.g., VBench) are publicly available. These details ensure that the experiments and results presented in this paper can be reproduced by the research community.

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

SUPPLEMENTARY MATERIAL: DUAL-IPO: DUAL-ITERATIVE PREFERENCE OPTIMIZATION FOR TEXT-TO-VIDEO GENERATION

# A    HUMAN ANNOTATION

Prior to the final manual annotation, we undertook several steps to ensure the quantity, quality, and efficiency of the annotation process. In building the dataset, we focused on the diversity and complexity of prompts, creating annotated data across multiple dimensions and categories. We meticulously selected candidates for the annotation task and provided them with in-depth training sessions. We then designed a user-friendly interface and a comprehensive annotation program to streamline the process. Additionally, we developed a detailed annotation guide that covers all aspects of the task and highlights essential precautions. To further ensure consistency and accuracy among annotators, we conducted multiple rounds of annotation and implemented random sampling quality checks.

## A.1    ANNOTATOR SELECTION

The effectiveness of annotated data hinges significantly on the capabilities of the individuals responsible for the annotation. As such, at the onset of the annotation initiative, we placed a strong emphasis on meticulously selecting and training a team of annotators to ensure their competence and objectivity. This process entailed administering a comprehensive evaluation to potential candidates, which was designed to gauge their proficiency in ten critical domains: domain expertise, tolerance for visually disturbing content, attention to detail, communication abilities, reliability, cultural and linguistic competence, technical skills, ethical awareness, aesthetic discernment, and motivation.

Given that the models under assessment could generate videos containing potentially distressing or inappropriate material, candidates were informed of this possibility in advance. Their participation in the evaluation was predicated on their acknowledgment of this condition, and they were assured of their right to withdraw at any time. Based on the results of the evaluation and the candidates' professional backgrounds, we sought to form a team that was diverse and well-rounded in terms of expertise and capabilities across the ten assessed areas. Ultimately, we selected a group of 10 annotators—comprising five men and five women, all of whom possessed bachelor's degrees. We conducted follow-up interviews with the selected annotators to reaffirm their suitability for the annotation task.

## A.2    ANNOTATION GUIDELINE

When it comes to assessing human preferences in videos, the evaluation process is structured around three fundamental aspects: **semantic consistency**, **motion smoothness**, and **video faithfulness**. Each of these dimensions is meticulously examined independently to provide a comprehensive understanding of the video's overall quality. **Semantic consistency** ensures that the video's content aligns logically and contextually, while **motion smoothness** evaluates the smoothness and naturalness of movements within the video. **Video faithfulness**, on the other hand, assesses how well the video maintains a realistic and high-quality appearance. By breaking down the evaluation into these distinct yet interconnected aspects, we can achieve a more nuanced and accurate assessment of human preferences in video content.

- **Semantic Consistency** The consistency between the text and the video refers to whether elements such as the topic category (e.g., people or animals), quantity, color, scene description, and style mentioned in the text align with what is displayed in the video.

- **Motion Smoothness** The motion smoothness refers to whether the movements depicted are smooth, coherent, and logically consistent, particularly in terms of their adherence to physical laws and real-world dynamics. This includes evaluating if the actions unfold in a seamless and believable manner, without abrupt or unrealistic transitions, and whether they align with the expected behavior of objects or individuals in the given context.

- **Video Faithfulness** Faithfulness refers to the accuracy and realism of the visual content in the video. It examines whether depictions of people, animals, or objects closely match their real-world counterparts, without unrealistic distortions like severed limbs, deformed features, or other anomalies. High fidelity ensures the visual elements are believable and consistent with reality, enhancing the video's authenticity..

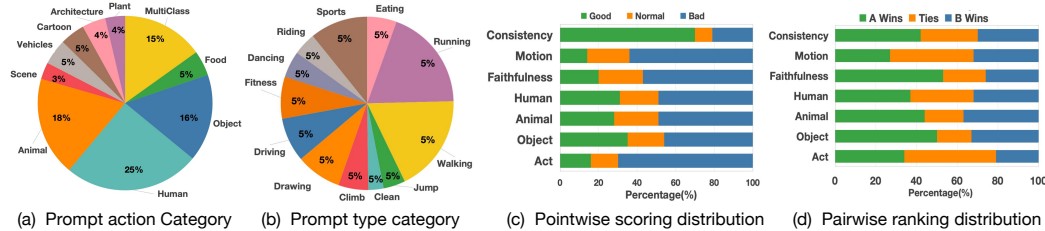

Figure A1: The distribution statistics of Human Preference Dataset on prompts in (a) and (b) as well as scoring/ranking in (c) and (d).

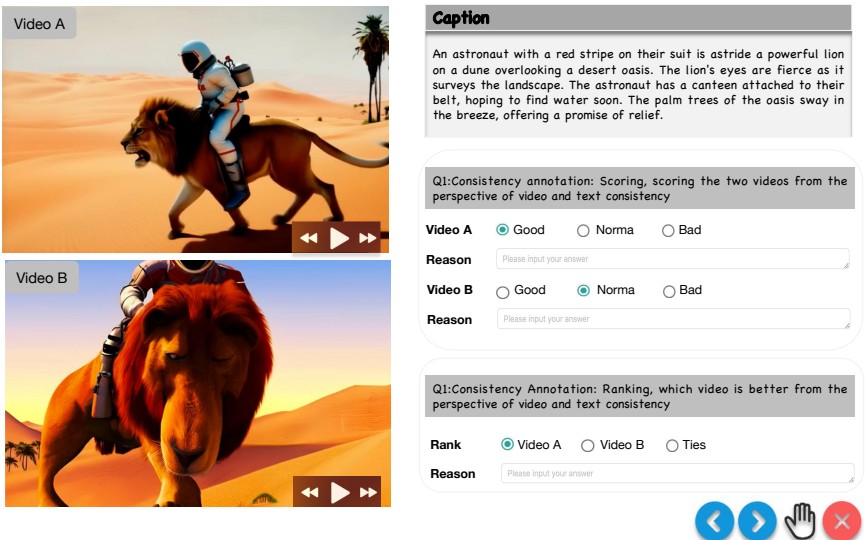

Figure A2: Our user interface presents one sample at a time to annotators, featuring four distinct icons for different functions.

**Pointwise Annotation**    For each dimension, annotators assign one of three ratings:

- **Good**: No observable flaws; fully satisfies the criterion.
- **Normal**: Minor imperfections that do not severely impact perception.
- **Bad**: Significant violations that degrade quality or realism.

Annotations are performed independently across dimensions to avoid bias.

**Pairwise Annotation**    Given two videos $\mathcal{V}_A$ and $\mathcal{V}_B$, annotators compare them under each dimension:

- For **semantic consistency**, determine which video better aligns with the intended context.
- For **motion smoothness**, identify the video with more natural and continuous motion.
- For **video faithfulness**, select the video exhibiting higher realism and fewer artifacts.

A ranked preference ($\mathcal{V}_A \succ \mathcal{V}_B$ or $\mathcal{V}_B \succ \mathcal{V}_A$) is recorded separately for each dimension. Ties are permitted only if differences are indistinguishable.

A.3    DATASET DISTRIBUTION

As shown in Fig. A1, the dataset includes over ten categories, such as Human, Architecture, Vehicles, and Animals, as well as more than ten different actions of human and animal. For each

prompt, the video generation model produces four variants with different random seeds. The dataset is iteratively expanded through an optimized regeneration strategy to improve preference alignment.

## A.4 ANNOTATION TRAINING AND INTERFACE

We convened a comprehensive training session centered on our detailed user guidelines to ensure that the annotation team fully comprehends our goals and standards. During the training, we covered the purpose, precautions, standards, workload, and remuneration related to the annotation process. Additionally, we formally communicated to the annotators that the human annotation is strictly for research purposes and that the annotated data might be made publicly available in the future. We reached a mutual agreement with the annotators regarding the standards, workload, remuneration, and intended use of the annotated data. The criteria for assessing video faithfulness, text-video alignment, and motion smoothness are universal, which means that individual standards should not vary significantly. Consequently, we annotated several sample videos using our carefully designed annotation platform to ensure consistency among annotators. An overview of the developed annotation platform is shown in A2. Through this training, we also equipped the annotators with the necessary knowledge to conduct impartial and detailed human evaluations of video faithfulness, text-video alignment, and motion smoothness.

| Model | Subject Consistency | Background Consistency | Temporal Flickering | Motion Smoothness | Dynamic Degree | Aesthetic Quality | Imaging Quality | Object Class |
|---|---|---|---|---|---|---|---|---|
| CogVideoX-2B | 96.78 | 96.63 | 98.89 | 97.73 | 59.86 | 60.82 | 61.68 | 83.37 |
| CogVideoX-2B-KTO$_1$ | 96.77 | 97.01 | 99.01 | 97.84 | 64.33 | 61.55 | 62 | 85.41 |
| CogVideoX-2B-KTO$_2$ | 96.81 | 97.23 | 99.21 | 98.03 | 68.76 | 62.35 | 61.77 | 85.63 |
| CogVideoX-2B-KTO$_3$ | 96.79 | 97.48 | 99.35 | 98.17 | 69.46 | 62.28 | 62.87 | 85.77 |

| Model | Multiple Objects | Human Action | Color | Spatial Relationship | Scene | Appearance Style | Temporal style | Overall Consistency |
|---|---|---|---|---|---|---|---|---|
| CogVideoX-2B | 62.63 | 98.00 | 79.41 | 69.90 | 51.14 | 24.8 | 24.36 | 26.66 |
| CogVideoX-2B-KTO$_1$ | 63.25 | 98.54 | 81.67 | 67.83 | 54.15 | 25.13 | 24.86 | 27.03 |
| CogVideoX-2B-KTO$_2$ | 63.44 | 98.97 | 83.24 | 70.16 | 54.35 | 25.47 | 26.44 | 27.37 |
| CogVideoX-2B-KTO$_3$ | 63.51 | 99.03 | 82.33 | 70.23 | 54.41 | 25.48 | 25.39 | 27.66 |

Table A1: Quantitative results on video assessment metrics. Display of comparison results between all indicators and baseline on vbench through multiple iterations.

## B ADDITIONAL QUANTITATIVE ANALYSIS

### B.1 ALL EVALUATION DIMENSION RESULTS

We present the evaluation metrics of VBench benchmark in 16 different dimensions in the Tab. A1. The results indicate that the performance shows a continuous upward trend in multiple training iterations, indicating that the training is dynamically stable. Although small fluctuations are observed, they can be ignored and will not affect the overall positive progress of the indicator.

| Model Scale | Pairwise Accuracy | Pointwise Accuracy |
|---|---|---|
| Dual-IPO-Reward-13B | 78.41 | 82.95 |
| Dual-IPO-Reward-40B | 81.33 | 85.57 |

Table A2: Accuracy of Reward Models on Validation Sets Across Different Scales. The reward models used in this study are finetuned from the ViLALin et al. (2023) multimodal large language model. This table evaluates their performance at different scales (13B and 40B) in terms of accuracy on pairwise and pointwise validation datasets. The results demonstrate that larger model scales yield higher accuracy for both pairwise ranking and pointwise scoring, indicating improved alignment with human preferences as the model size increases.

### B.2 IMPACT OF REWARD MODEL SCALE ON ACCURACY

To evaluate the impact of model scale on reward performance, we experimented with two ViLA-based critic models of 13B and 40B parameters. Fine-tuned on paired and pointwise annotated datasets, these models were assessed on validation tasks, as shown in Tab. A2.

Results show that increasing the model size from 13B to 40B significantly improves accuracy, with the 40B model achieving 80.73% in paired ranking and 86.57% in pointwise scoring—2.92% and

2.62% higher than the 13B model. This suggests that larger critic models better capture human preferences by representing complex multimodal relationships.

Our ablation study further highlights the importance of model scale. While smaller models perform reasonably well, larger models significantly boost accuracy, benefiting downstream reinforcement learning tasks. These findings suggest that scaling the critic model is a simple yet effective way to enhance preference alignment and improve video generation quality.

| Models | Total Score | Quality Score | Semantic Score | Subject Consist. | Background Consist. |
|---|---|---|---|---|---|
| CogVideoX-2B | 80.91 | 82.18 | 75.83 | 96.78 | 96.63 |
| w/IPO-Reward-13B | 82.42 | 83.53 | 77.97 | 96.81 | 97.23 |
| w/IPO-Reward-40B | 82.52 | 83.63 | 78.07 | 96.83 | 97.37 |

| Models | Temporal Flicker | Motion Smooth. | Aesthetic Quality | Dynamic Degree | Image Quality |
|---|---|---|---|---|---|
| CogVideoX-2B | 98.89 | 97.73 | 60.82 | 59.86 | 61.68 |
| w/IPO-Reward-13B | 99.21 | 98.03 | 62.35 | 68.76 | 61.77 |
| w/IPO-Reward-40B | 99.19 | 98.09 | 62.38 | 68.92 | 62.01 |

Table A3: Evaluate the impact of the critic model on performance, verify the performance of the critic model results for different model sizes of 13B and 40B, a more accurate critic model leads to improved results.

## C  HUMAN-VALIDATED SRPO AND RELIABILITY OF SYNTHETIC PREFERENCES

SRPO relies on pseudo-preference labels generated by the critic, which raises the question of how to ensure that these synthetic labels remain aligned with human judgments. To address this, our implementation couples *every* SRPO iteration with an explicit human-validation step on a held-out preference test set, and we only accept critic updates that pass this validation.

**Human preference test set.**  We maintain a fixed human-labeled preference test set constructed from the current T2V model. For each prompt, the model generates multiple candidate videos, and annotators rank these candidates. The resulting rankings are converted into pairwise preference labels and used solely for evaluation, not for training SRPO. This set allows us to measure how well the critic agrees with human judgments on realistic model outputs.

**SRPO iteration and validation loop.**  Each SRPO iteration proceeds as follows (see Figure A3):

1. **Pseudo-label generation with PCE filtering.** Given the current critic, we score candidate video pairs generated by the T2V model and retain only those with high confidence according to a prediction-confidence estimator (PCE). These high-confidence pairs form the synthetic preference set.

2. **Critic update.** The critic is fine-tuned on a mixture of the original human-labeled seed data and the filtered pseudo-labeled pairs, yielding an updated reward model.

3. **Human-validation step.** After the update, we evaluate the critic on the human preference test set and compute the agreement between the critic's predicted preferences and human rankings. On this same set, we also evaluate open-source reward models as baselines and report the resulting human agreement in Table 1. In Table 6, we track both this agreement and downstream VBench metrics before and after SRPO updates.

4. **Acceptance criterion.** In practice, we only accept an SRPO update if the critic's agreement with human labels remains above a fixed threshold (75% in our experiments). In addition, we monitor VBench metrics of the aligned generator; if these metrics degrade after an SRPO update, we discard the update and retrain or roll back the critic. This mechanism

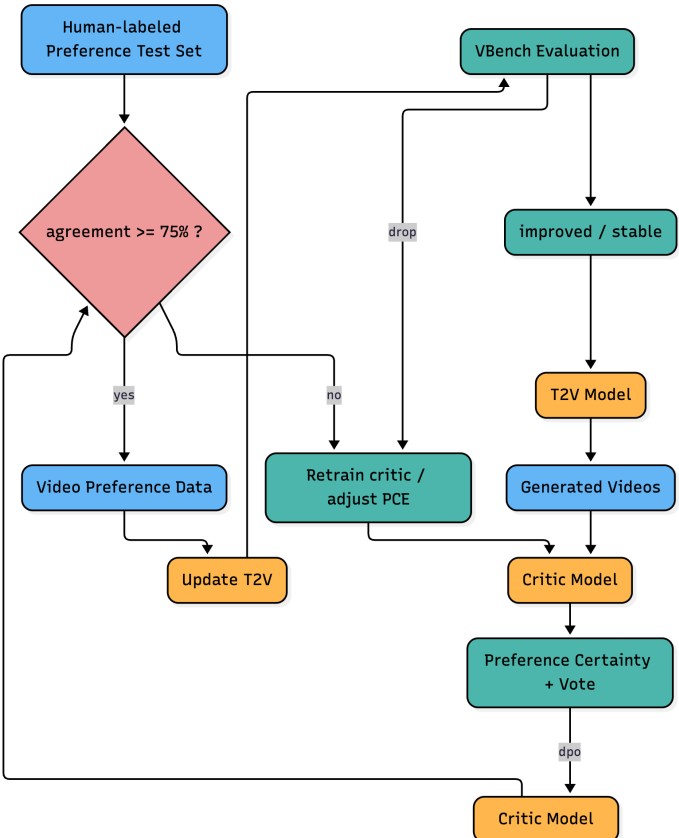

Figure A3: Overview of the SRPO iteration and validation loop. High-confidence pseudo-labels are generated via PCE filtering, used to update the critic, and each update is accepted only if it passes a human-agreement threshold and maintains or improves downstream VBench performance.

prevents error accumulation from low-quality pseudo-labels and ensures that SRPO steps are beneficial both for the reward model and for downstream video quality.

This human-in-the-loop validation makes the SRPO refinement process both controllable and auditable: synthetic labels are used aggressively to scale supervision, but each refinement step is gated by human-validated agreement and downstream metrics, which mitigates the risk of drift away from true human preferences.

## D    NUMBER OF DUAL-IPO ROUNDS AND STOPPING CRITERIA

Dual-IPO is designed as a multi-round optimization procedure in which the generator and critic are updated iteratively. An important practical question is how many rounds are beneficial, and when the procedure should be stopped to avoid diminishing returns or overfitting. In this section, we summarize our empirical observations and the resulting stopping strategy.

**Empirical saturation of VBench performance.**    Beyond the rounds reported in the main experiments, we conducted an additional Dual-IPO iteration on Wan-1.3B. We observed that the overall VBench score had essentially saturated: the score only changed from 88.32 to 88.33 in this extra round. Similar saturation phenomena appear on other configurations. In contrast, our internal reward continued to increase. This indicates that performance does not keep improving indefinitely with more Dual-IPO iterations, and that after a certain point further optimization mainly improves the internal reward signal rather than the external evaluation metrics.

**Discrepancy between internal reward and external metrics.** On VBench's 16 dimensions, our aligned models already achieve very high scores after a moderate number of rounds; the remaining headroom is concentrated in aesthetics and visual quality. In later iterations, the internal critic reward keeps increasing, but this no longer translates into measurable gains on VBench. One could potentially reweight or redesign the reward dimensions to more aggressively target the remaining VBench dimensions and obtain higher benchmark scores, but this would amount to tuning the reward directly to the benchmark, which risks drifting toward "benchmark hacking" rather than genuine quality improvements.

**Practical early stopping rule.** Given these observations, we do not aim to identify a globally optimal number of Dual-IPO rounds. Instead, we adopt a practical early stopping rule: we stop the iterative procedure once VBench and human evaluation no longer show meaningful improvements, even if the internal reward continues to rise. Concretely, we monitor:

- the overall and per-dimension VBench scores,
- human preference evaluations on held-out comparison sets,
- the critic's agreement with human labels.

When these external signals saturate or begin to fluctuate within a narrow range, additional rounds are deemed unproductive and the current model is selected.

**Limitations and future exploration.** Our experiments suggest that Dual-IPO provides a practically useful recipe for multi-round optimization with reasonable stopping criteria: it yields consistent gains over the baseline without inducing model collapse, severe reward hacking, or overfitting to a particular critic. Nonetheless, we do not claim to have identified a universally optimal configuration. A truly complete "optimal recipe" will depend on multiple engineering factors, including model size, the number of rounds, the number of samples per round, and the choice of evaluation metrics. A more systematic study of these scaling behaviours, especially with additional metrics beyond VBench and on larger models, is an important direction for future work.

## E    THEORETICAL MOTIVATION AND CONVERGENCE ANALYSIS

Dual-IPO adopts a dual-iterative optimization scheme in which the policy (generator) and the critic (reward model) are updated in alternating stages. This design is supported by existing theoretical and empirical findings from iterative RLHF-style training, as well as by standard KL-regularized RL objectives used in DPO-like analyses.

**Motivation from iterative RLHF.** Prior work on iterative RLHF training Touvron et al. (2023a;b) shows that (i) updating the reward model during training is crucial for maintaining alignment with the current policy, and (ii) using data labeled by the current policy to refine the reward model can significantly improve its accuracy and robustness. In these pipelines, the policy and reward model are updated in alternating stages rather than in a single offline pass. Dual-IPO follows the same principle: our SRPO component refines the critic using self-labeled preference data, filtered by a certainty estimator to reduce noise. As the generator improves, the critic is iteratively updated on higher-quality, policy-generated data, which gradually tightens the alignment between the learned reward and the target preference signal.

**Connection to KL-regularized RL and DPO.** The effectiveness of iterative policy updates in Dual-IPO is further motivated by the standard KL-regularized RL objective underlying DPO-like methods Zhang (2023); Pang et al. (2024). The optimal aligned policy for a given reward function $r(x_0, c)$ can be written as

$$\pi^*(x_0 \mid c) \propto \pi_0(x_0 \mid c) \exp\left(\frac{1}{\beta} r(x_0, c)\right), \tag{A1}$$

where $\pi_0$ is a reference policy and $\beta > 0$ is a temperature parameter. Online RL methods such as PPO and GRPO can be viewed as stochastic optimization procedures that (approximately) move $\pi_0$ toward $\pi^*$ under online data collection. Empirically, such iterative procedures often outperform a

single offline DPO step. This motivates our iterative optimization scheme: by repeatedly updating the generator with DPO/KTO under an increasingly accurate critic, we expect the current policy to gradually approach $\pi^*$, leading to more stable and higher-quality solutions than a one-shot optimization.

**Practical convergence and stopping criterion.** In practice, we observe that performance metrics (e.g., VBench scores) and training losses improve over successive Dual-IPO iterations and then saturate. We exploit this behaviour to define a practical stopping rule: we terminate the iterative process once additional Dual-IPO iterations bring only marginal improvements on VBench. We attribute this saturation partly to the current reward design and the coverage of existing video-generation metrics, as well as to potential capacity limits of smaller models. Although resource constraints prevent running arbitrarily long iteration sweeps, extended runs consistently show a pattern of improvement followed by saturation rather than divergence.

**Stability in training.** Mild oscillations in losses and metrics are inevitable in reinforcement-learning-style training, and Dual-IPO is no exception. In our context, *stability* refers to controlled behaviour of the system rather than perfectly monotone curves. Concretely, in each Dual-IPO iteration we monitor VBench scores and training losses; when signs of degradation or instability are detected (e.g., VBench scores drop or losses behave pathologically), we trigger a new Dual-IPO iteration and refine the critic via SRPO. Across all reported experiments, the metrics exhibit consistent improvement followed by saturation, without wild oscillations or runaway divergence. We provide the corresponding curves, including later-iteration results, to empirically illustrate this convergence pattern.

Overall, Dual-IPO is theoretically motivated by KL-regularized RL and iterative RLHF, and empirically exhibits stable, saturating behaviour under our monitoring and stopping criteria.

## F PROMPT CONSTRUCTION PIPELINE

We design a programmatic prompt construction pipeline that combines structured templates, large vocabularies, and LLM-based expansion and filtering. The goal is to obtain an effectively unbounded prompt pool that is (i) large and diverse, (ii) balanced across high-level categories, and (iii) explicitly designed to mitigate cultural and demographic biases. Figure 1 in the appendix summarizes the distributions of major subject and action sub-categories.

### F.1 DESIGN GOALS

Our prompt pipeline is built around three main desiderata:

- **Size.** Prompts are generated from templates with random instantiation rather than from a fixed list, so the space of possible prompts is effectively unbounded and exact repetitions are rare.
- **Diversity.** The templates factorize subjects, attributes, spatial relations, actions, and scene contexts, which are populated from rich vocabularies and combined into both simple and complex scenes.
- **Balance and bias awareness.** We impose explicit sampling distributions over high-level categories (e.g., humans vs. animals vs. objects; single vs. multiple subjects; easy vs. difficult actions) and design the subject vocabularies and filtering stages to reduce systematic underrepresentation of particular demographic or semantic groups.

### F.2 STRUCTURED PROMPT SPACE

**Subjects and attributes.** We organize subjects into a structured space with two primary groups, humans and animals, and additional object categories.

- **Human subjects.** Humans are enumerated over age (child, adult, elderly), gender-neutral and gendered roles (e.g., teacher, student, firefighter, musician), and physical attributes

such as height, body shape, and hair style/color. Forms of address vary from generic ("a young person") to role-specific ("an elderly firefighter").

- **Animal subjects.** Animals are divided into land, flying, and marine species, with a large set of concrete types and surface forms (e.g., "golden retriever", "tabby cat", "bald eagle").
- **Additional object categories.** We include vehicles, food, and common objects (e.g., furniture, tools, electronic devices, toys). These appear as primary subjects or as contextual elements interacting with humans and animals.

Subjects can appear as single targets or as multiple interacting entities. For multi-subject prompts, we additionally sample explicit spatial and relational patterns, such as left/right, in front of/behind, above/below, or role-based relations (e.g., "a child standing next to a dog", "two cyclists racing behind a car"). This encourages the model to reason about structured multi-object configurations.

**Action taxonomy.** To better cover temporal behavior, we categorize actions into three difficulty levels:

- **Simple** (e.g., walking, running, sitting, waving),
- **Medium** (e.g., jumping, crawling, turning around, picking up objects),
- **Difficult** (e.g., complex sports or task-oriented behaviors such as gymnastics or acrobatic moves).

Actions are combined with subjects and then filtered. Non-realistic pairs (e.g., an inanimate object performing a complex motion) are either discarded or explicitly mapped to stylized domains ("cartoon", "dreamlike", "sci-fi"), so that even unrealistic combinations become coherent abstract scenes.

**Scene contexts.** We define many scene types grouped into realistic and abstract contexts:

- **Realistic scenes** include common natural and urban environments such as forests, snowfields, farmland, parks, kitchens, offices, and city streets.
- **Abstract and surreal scenes** include deliberately unusual compositions, such as "a car driving inside a coffee cup" or "a cat floating through a neon-colored dreamscape". These are used to create a long tail of challenging prompts without dominating the dataset.

### F.3    SAMPLING STRATEGY AND TARGET DISTRIBUTIONS

We do not fix a finite list of prompts per category. Instead, each training run defines a *target distribution* over high-level prompt categories (e.g., humans vs. animals vs. objects, single vs. multiple subjects, easy vs. harder actions, realistic vs. abstract scenes). Prompts are then sampled on the fly from the global structured pool under this distribution.

Because subjects and attributes are drawn from broad, approximately uniform vocabularies, individual prompts almost never repeat and no single subject type is systematically underrepresented. At the same time, the category-wise sampling weights ensure that each epoch sees a roughly balanced mixture of content types rather than being dominated by a single class.

We also stratify actions by difficulty and choose their sampling probabilities according to the capability of the base model (for example, placing more mass on simple actions for smaller backbones). Overall, we adopt a schedule in which the sampling probability gradually decreases from:

- simple to difficult actions,
- realistic to abstract scenes,
- single-subject to multi-subject compositions,
- everyday to highly surreal situations.

This keeps the dataset dominated by realistic, common scenarios while still maintaining a diverse long tail of challenging compositions involving multiple subjects, attributes, spatial relations, and abstract scenes.

### F.4 LLM-Based Expansion and Safety Filtering

Textual prompts can themselves be a source of cultural and demographic bias, so we deliberately restrict and structure how we use large language models in the pipeline.

**Restricted use of the LLM.** The core discrete choices—subject category, attributes, actions, and scene type—are sampled directly from our structured, enumerated pools. The LLM is only used to:

1. expand these discrete tuples into fluent natural-language descriptions, and
2. detect and repair problematic prompts.

Crucially, the LLM does not decide which demographic or object type appears; those decisions are controlled by our explicit vocabularies and sampling weights.

**Automatic filtering and auditing.** We apply multiple LLM-based filtering stages to remove or rewrite prompts that are:

- nonsensical or internally inconsistent,
- unsafe (e.g., self-harm, explicit violence),
- clearly discriminatory, offensive, or otherwise inappropriate.

Prompt–caption pairs that are flagged as problematic are discarded or rewritten into safe, abstract variants. In addition, we perform light manual spot checks on random batches and monitor empirical distributions over subject and action categories during training to keep coverage roughly balanced. This design makes the prompt pool both controllable and auditable, and it provides clearer levers for mitigating bias than relying on scraped, uncurated prompts.

### F.5 Bias Mitigation and Limitations

Our construction incorporates several mechanisms aimed at reducing cultural and demographic bias:

- **Enumerated human subject space** with explicit variation over age, roles, and physical attributes, sampled approximately uniformly.
- **Balanced high-level categories** controlled by sampling weights for humans, animals, and objects, as well as single vs. multiple subjects and action difficulty.
- **Restricted LLM usage** so that demographic composition is determined by transparent, discrete pools rather than opaque generative behavior.
- **Automatic and manual filtering** for unsafe or discriminatory content, coupled with monitoring of empirical distributions.

These measures do not eliminate bias entirely, but they make the prompt space more controllable and inspectable, and allow us to balance demographic and semantic coverage more effectively than using unfiltered real-world text prompts. A more systematic fairness and bias analysis of downstream models trained under this prompt distribution remains an important direction for future work.

### F.6 Reproducibility

We will release the main prompt templates, vocabularies, and sampling configurations used in our pipeline to facilitate reproducibility and future research. This will allow others to regenerate comparable prompt distributions or adapt our structured construction to new domains and models.

## G Computational Cost and Practicality of Dual-IPO

We thank the reviewer for raising the issue of computational cost and scalability. Our current experiments indeed use a relatively large configuration in order to push performance and demonstrate

the full potential of Dual-IPO, and we explicitly list this as a limitation. At the same time, Dual-IPO is designed to be *small-lab friendly*: it reduces the dependence on repeated human annotation, keeps the additional optimization overhead modest and largely reusable, and attains strong (often SOTA-level) performance starting from comparatively small generators. In this section, we clarify (i) where the main computational bottlenecks lie, (ii) in what sense Dual-IPO is resource-friendly, and (iii) what practical trade-offs and cost-reduction strategies are available.

## G.1    Where the Compute Actually Goes

In our current system, the dominant practical bottleneck comes from **video generation and evaluation**, rather than from the Dual-IPO algorithm itself:

- **Video sampling.** Generating large numbers of videos from a high-quality T2V model is intrinsically expensive and tends to dominate wall-clock time in any RL- or preference-based alignment pipeline for video (similar observations have also been made in prior work such as DanceGRPO).
- **Automatic evaluation.** Running VBench and other automated evaluators on the generated videos adds another substantial cost, which is again shared by essentially all strong alignment methods that rely on large-scale offline evaluation.

By comparison, the additional overhead introduced by Dual-IPO itself is relatively small. In our implementation, a single SRPO-based critic update takes roughly one day on $8\times$A100 GPUs. Once a small seed set of human annotations is prepared, all later reward-model updates are fully automated and do not require additional human labeling. Thus, most of the overall cost is *shared* with any competitive T2V alignment pipeline, whereas the incremental cost of Dual-IPO is modest.

## G.2    Comparison to Standard RLHF Pipelines

A key motivation for Dual-IPO is to avoid the continuous human annotation that typically constitutes the main bottleneck in RLHF-style pipelines. In many RLHF setups for LLaMA-style models [1,2], the reward model is repeatedly updated on fresh model outputs using large-scale human preference datasets, and each new RL round requires collecting additional human labels. This process is:

- time- and labor-intensive,
- difficult and expensive to reproduce for typical academic or small industrial labs,
- tightly coupled to large-scale human preference collection infrastructure.

In contrast, Dual-IPO is designed to keep *human supervision* small and front-loaded:

- We start from a small amount of human-labeled data to seed the critic.
- We then apply SRPO to generate pseudo-labels and perform self-refinement of the critic, *without* further human annotation.
- One SRPO-based critic update is a single, bounded procedure (about one day on $8\times$A100 in our configuration), after which the critic can be re-used across multiple generators or training runs.

Compared with previous approaches that rely on tens of thousands to hundreds of thousands of manually labeled preference pairs and repeated reward-model training, our main focus is to *reduce human annotation cost* while still allowing the reward model to be updated during training. From the perspective of annotation and maintenance cost, this makes Dual-IPO relatively lightweight and more practical for small research groups.

## G.3    Generator and Critic Configurations

Dual-IPO is explicitly designed to reach strong performance starting from comparatively small generators, with a critic stage that is modest relative to full T2V pretraining.

**Generators.** Our main gains come from aligning a 2B CogVideoX model, which achieves SOTA-level VBench performance and surpasses a 5B baseline after only three Dual-IPO iterations. This is a model scale much closer to what typical research labs can afford, and demonstrates that substantial alignment benefits can be obtained without training or fine-tuning extremely large generators.

**Critics.** On the critic side, we fine-tune a pre-trained VILA (13B/40B) reward model for 5 epochs on 16 video frames with a small batch size on 32 GPUs, and then apply SRPO to generate pseudo-labels for further refinement. This is a modest post-training stage compared to full T2V pretraining, and the resulting critic can be *reused* across multiple generators or experiments. Labs with more limited resources can:

- use smaller critic models,
- reduce the number of SRPO iterations,
- or train on fewer frames per video

to trade off performance against cost in a controlled and transparent way.

### G.4 COST-REDUCTION STRATEGIES AND TRADE-OFFS

The configuration we report in the main paper should be viewed as a *best-practice* setting aimed at pushing SOTA performance, not as a minimum requirement of the algorithm. In practical deployments, there are several trade-offs that can substantially reduce computational cost:

- **Smaller T2V models.** As shown by the CogVideoX-2B experiments, smaller base generators can already benefit strongly from Dual-IPO and even surpass larger baselines after a few iterations.
- **Smaller critics.** A smaller reward model, or even a partially frozen backbone with lighter adaptation, can still provide useful gradients for preference optimization.
- **Fewer samples and iterations.** Reducing the number of generated videos per iteration and/or the total number of Dual-IPO iterations directly scales down both generation and evaluation cost, while still yielding noticeable performance gains.
- **Reusing critics across runs.** Once a critic has been trained with SRPO, it can be reused across different generators and experiments, amortizing its cost over multiple projects.

In all of these settings, the main wall-clock time remains dominated by video generation and evaluation, which any competitive preference-based alignment method must pay. Dual-IPO adds a manageable, well-structured optimization layer on top of this shared cost, rather than introducing fundamentally new bottlenecks.

### G.5 SUMMARY

In summary, Dual-IPO is designed with both *computational efficiency* and *practical deployability* in mind:

- The dominant cost arises from video sampling and evaluation, which is common to strong RL-based video training pipelines.
- Dual-IPO significantly reduces human annotation requirements compared to standard RLHF pipelines by using a small seed of human labels together with SRPO-based pseudo-labeling and critic self-refinement.
- The algorithm achieves strong, often SOTA-level performance starting from a 2B generator and a modest critic post-training stage, which are more accessible to typical research labs.
- Multiple knobs (model size, number of iterations, samples per iteration, critic capacity) allow practitioners to flexibly trade off performance against cost.

We therefore believe that, while our reported configuration is computationally demanding, Dual-IPO as a framework remains relatively lightweight and scalable, and is suitable for small and medium-sized labs that seek strong alignment performance without large-scale, ongoing human annotation.

## G.6 Impact of Reward Model Size on RL

We conducted a comparative analysis of reward models of different sizes in the iterative preference optimization training process. As shown in Tab. A3, larger reward models provide more precise reward estimations, leading to improved alignment with human preferences. This enhanced reward accuracy contributes to better optimization during preference alignment training, ultimately resulting in superior model performance. Our findings highlight the importance of scaling the critic model to achieve more reliable preference optimization and reinforce the benefits of leveraging larger critic architectures for improved alignment outcomes.

## H Additional Qualitative Comparison

We provide a more detailed qualitative comparison in Fig. A4 and Fig. A5, which demonstrates the superiority of our model over the baseline. This visualization highlights key improvements in fidelity and overall quality.By comparing outputs side by side, we illustrate the advantages of our model in handling complex scenarios, reinforcing its effectiveness in real-world applications.

## I Use of LLM

Large Language Models (LLMs) were employed solely as auxiliary tools to improve the efficiency of manuscript preparation. Their use included assisting in retrieving related literature, editing and formatting LaTeX mathematical expressions, polishing the language style, debugging minor code or visualization scripts, and formatting references.

## J Societal Impacts

The emergence of advanced text-to-video generation pipelines and optimized training strategies marks a major breakthrough in AI-driven content creation. These innovations significantly lower barriers to video production, making it more efficient, cost-effective, and widely accessible. Industries such as education, entertainment, marketing, and accessibility technology stand to benefit immensely, as creators can generate high-quality videos with minimal effort and resources.

However, alongside these advancements come ethical concerns, particularly regarding potential misuse. One of the most pressing risks is the creation of highly realistic yet misleading content, such as deepfakes, which could be exploited for misinformation, reputational harm, or social manipulation. Additionally, automated video curation may unintentionally reinforce biases present in training datasets, leading to the perpetuation of stereotypes or discriminatory narratives.

On a larger scale, the widespread adoption of AI-driven video generation could disrupt traditional media industries, impacting professions like video editing, animation, and scriptwriting. The proliferation of hyper-realistic AI-generated content also raises concerns about media authenticity, making it increasingly difficult for audiences to distinguish between real and synthetic footage. Furthermore, if access to these technologies remains limited to well-funded organizations, existing digital divides could widen, leaving smaller creators at a disadvantage.

To mitigate these risks, proactive measures must be taken. Implementing transparency features—such as metadata labels to distinguish AI-generated content—can enhance accountability. Ensuring diverse and ethically curated datasets is crucial to reducing bias and promoting fairness in outputs. Policymakers, industry leaders, and researchers must work together to develop regulatory frameworks that balance innovation with ethical responsibility. Additionally, providing educational resources and affordable access to AI tools can help democratize their benefits, preventing technological disparities.

The rapid evolution of generative AI necessitates ongoing ethical oversight. Continuous evaluation of its societal impact, coupled with adaptive safeguards, is essential to minimizing harm while maximizing positive outcomes. Addressing these challenges requires a multidisciplinary approach, fostering a responsible AI ecosystem that empowers creators without compromising ethical integrity.

# K LIMITATIONS

While our approach represents a significant advancement in text-to-video generation, several challenges persist. Addressing these issues presents valuable opportunities for future research and development. Below, we outline key limitations and potential solutions:

**Computational Constraints** Despite improvements in training efficiency, large-scale text-to-video models still demand substantial computational power. This limitation may hinder accessibility, particularly for individuals and organizations with limited resources. Future research should focus on developing lightweight architectures, model compression techniques, and efficient inference strategies to enhance accessibility in resource-constrained environments.

**Dataset Representativeness and Cultural Adaptability** The effectiveness of our model is highly dependent on the quality and diversity of its training dataset. While significant efforts were made to curate a representative dataset, certain cultural contexts, niche topics, and underrepresented communities may not be adequately covered. Additionally, biases present in pretraining data could lead to unintended stereotypes or inappropriate outputs. Expanding dataset coverage, improving curation techniques, and integrating fairness-aware training strategies will be crucial to enhancing contextual accuracy and mitigating bias.

**Multimodal Hallucination and Interpretability** Since our evaluation models are fine-tuned multimodal large language models (MLLMs), they are susceptible to hallucination—generating content that appears plausible but is factually incorrect or cannot be inferred from the input text and images. Furthermore, MLLMs inherit biases from their pretraining data, which may lead to inappropriate responses. While careful curation of supervised fine-tuning (SFT) data helps alleviate these issues, they are not fully resolved. In addition to these concerns, MLLMs also suffer from opacity, interpretability challenges, and sensitivity to input formatting, making it difficult to understand and control their decision-making processes. Enhancing model explainability and robustness remains a crucial area for future research.

**Human Annotation Limitations** Human evaluation plays a critical role in assessing model performance, but it is inherently subjective and influenced by annotator biases, perspectives, and inconsistencies. Errors or noisy labels may arise during annotation, potentially affecting evaluation reliability. To address this, we conducted multiple rounds of trial annotations and random sampling quality inspections to improve inter-annotator consistency. Additionally, we designed user-friendly annotation guidelines and interfaces to streamline the process and enhance accuracy. However, variations in annotation methodologies across different platforms and annotators can still lead to discrepancies, limiting the generalizability of our results. Moreover, human annotation remains time-consuming and resource-intensive, constraining scalability. Future efforts could explore semi-automated evaluation approaches or crowdsourced methods to improve efficiency while maintaining quality.

**Ethical Considerations and Regulatory Compliance** Ensuring responsible AI deployment remains an ongoing challenge, even with safeguards in place to prevent misuse. The rapid advancement of AI-generated media demands continuous revisions to ethical policies and regulatory frameworks. Proactively engaging with policymakers, industry leaders, and researchers is crucial for developing guidelines that uphold accountability, transparency, and fair usage. Embedding metadata or digital watermarks in AI-generated content can improve traceability and help differentiate synthetic media from authentic sources. Furthermore, raising public awareness and education on AI-generated content help users recognize potential risks. Developers should prioritize responsible AI innovation by ensuring algorithmic fairness while minimizing bias. Additionally, fostering cross-industry collaboration and promoting global standards will contribute to a more secure and trustworthy AI ecosystem.

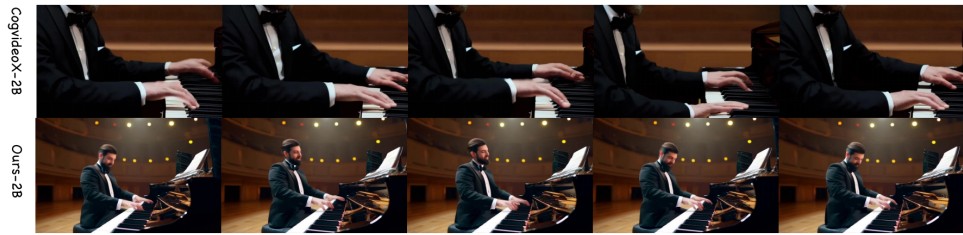

A man with a beard is playing the piano in a concert hall. He wears a black tuxedo and a bow tie. His fingers dance on the keys as he plays a beautiful melody.

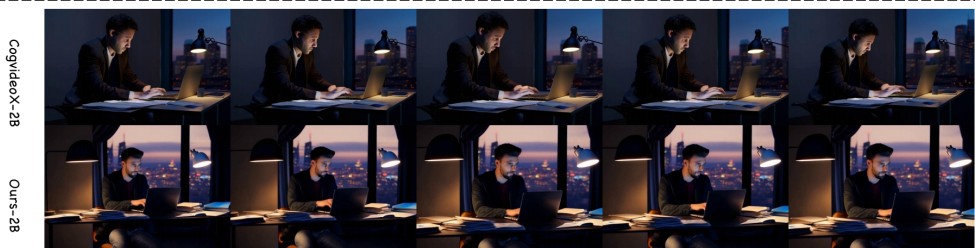

A director, dressed in a dark blazer and casual jeans, sits at a modern desk cluttered with scripts and notes. He types intently on a sleek laptop, his brow furrowed in concentration. The room is dimly lit by a single desk lamp, casting a warm glow over his focused expression. Behind him, a large window frames a city skyline at dusk, adding a sense of urgency and creativity to the scene.

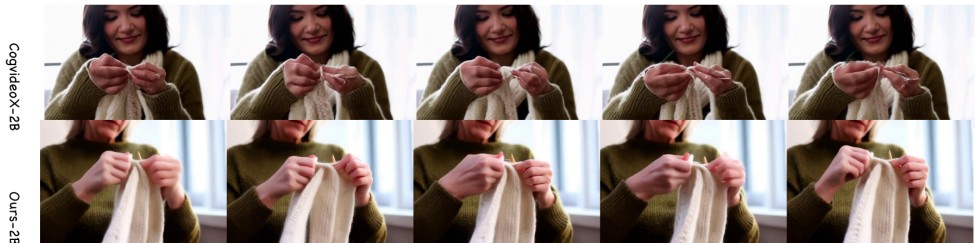

A woman sits in a cozy, well-lit room, her hands deftly working a long, creamy white scarf. She wears a warm, olive-green sweater and a gentle smile, her eyes focused on the intricate pattern of her knitting. The soft, natural light streaming through a nearby window highlights the texture of the yarn and the calm, serene atmosphere of the space. Her concentration and skill are evident as she effortlessly manipulates the needles, creating a beautiful, handmade scarf.

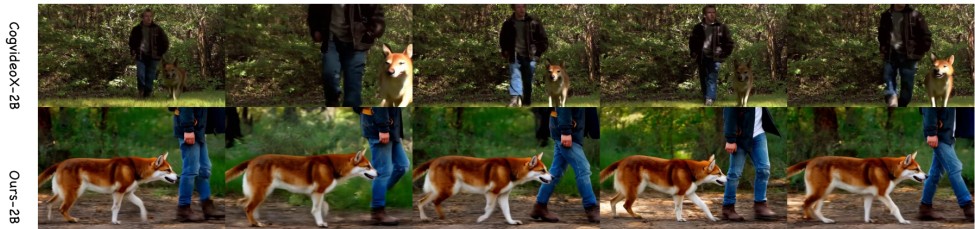

An actor dressed in rugged, casual attire, including a dark jacket, jeans, and hiking boots, walks alongside a dingo in a natural, wooded setting. The dingo, with its distinctive reddish-brown fur and alert eyes, moves gracefully beside the actor, who looks calm and engaged. The background features lush greenery and dappled sunlight filtering through the trees, creating a serene and harmonious atmosphere. The actor and the dingo walk in sync, their bond evident in their synchronized steps and mutual awareness of each other.

Figure A4: **Enhanced qualitative comparison**. We conduct a qualitative comparison between CogVideo-2B and Dual-IPO-2B to evaluate their performance

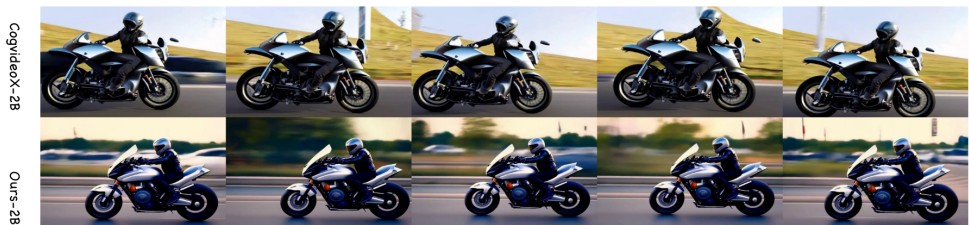

A silver flying motorcycle with a sleek design and a powerful engine zooms through a city. The motorcycle's rider wears a helmet and leather jacket as they navigate through traffic.

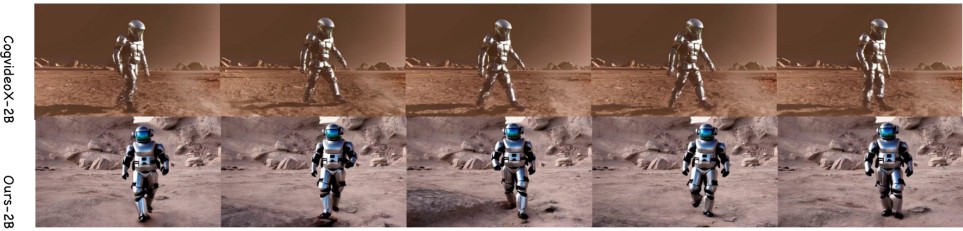

A man dressed as a robot astronaut walks across a barren, rocky landscape on a distant planet. His sleek, metallic suit glimmers under the dim light of a distant sun, and his helmet reflects the alien terrain. He strides purposefully, leaving faint footprints in the dust, with a sense of determination and exploration. The desolate environment, dotted with boulders and sparse vegetation, enhances the otherworldly atmosphere of his journey.

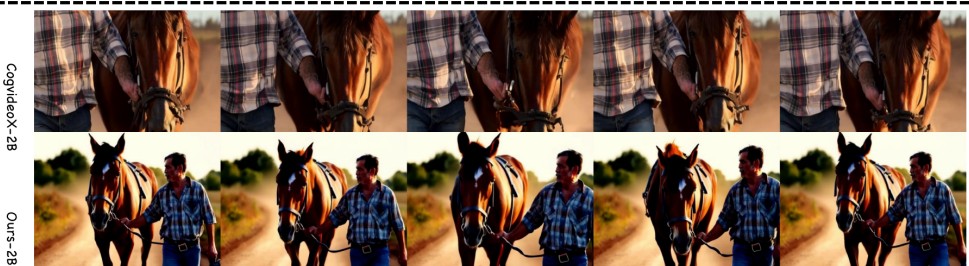

A rugged worker in a plaid shirt and denim jeans leads a strong, chestnut-colored horse along a dusty, rural path. The horse, adorned with a simple leather harness, walks steadily beside the worker, who holds the reins with a firm yet gentle grip. The scene is set in the late afternoon, with warm sunlight casting long shadows and highlighting the textures of the dirt road and the surrounding countryside. The worker's weathered face and the horse's powerful presence convey a sense of steadfast determination and mutual respect.

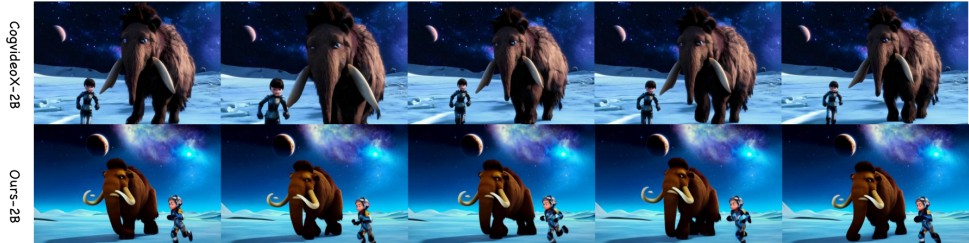

A young girl in a futuristic spacesuit runs across a vast, icy landscape on a distant planet, with a towering mammoth-like creature beside her. The mammoth's massive, shaggy form and long tusks contrast with the stark, alien environment. The sky above is a deep, star-filled space, with distant planets and nebulae visible. The girl's determined expression and the mammoth's steady pace convey a sense of adventure and discovery in this otherworldly setting.

Figure A5: **Enhanced qualitative comparison**. We conduct a qualitative comparison between CogVideo-2B and Dual-IPO-2B to evaluate their performance

