# OpenReview forum: "Dual-IPO: Dual-Iterative Preference Optimization for Text-to-Video Generation"
_ICLR.cc/2026/Conference — ICLR 2026 Poster_

### Official Review · Reviewer_gKUH · 2025-10-30

**Soundness:** 3
**Presentation:** 3
**Contribution:** 3
**Rating:** 6
**Confidence:** 4

**Summary:**

The paper introduces Dual-Iterative Preference Optimization (Dual-IPO), a framework that jointly and repeatedly improves both the reward model and the video generation model to better align diffusion-based T2V models with human preferences. The reward model is strengthened in each round via Chain-of-Thought guided reasoning, voting-based self-consistency, and a preference-certainty estimator to produce reliable pseudo-labels without extensive manual annotation. Using these refined signals, the video generator is iteratively optimized via DPO/KTO-style preference learning, and both models co-evolve over multiple rounds. Experiments across architectures and model sizes show consistent gains in subject consistency, motion smoothness, and aesthetics, including cases where a 2B model surpasses a 5B baseline.

**Strengths:**

- Dual-IPO attains a level of preference alignment that was not achievable with prior approaches. It is validated on state-of-the-art backbones (CogVideoX, Wan 2.1), and even a small model is shown to surpass the performance of a larger one in Table 5.
- The effectiveness of the constructed reward models is discussed, particularly centered around Table 1.
- Human evaluation results (e.g., Fig. 3b) are also included for complementary aspects, demonstrating that the improvements are not merely due to over-optimization against the evaluation model.

**Weaknesses:**

- The discussion on the number of iterative rounds is insufficient. It is unclear whether increasing the number of rounds continues to improve performance or saturates, and therefore, it is difficult to claim that an optimal recipe has been established.

- For the experiments on Wan, not all VBench metrics are reported. Since VBench is biased toward temporal consistency, it remains possible that the training procedure reduces temporal dynamics in videos. This should be explicitly addressed. The trade-off between temporal consistency and dynamics has been discussed in prior work (e.g., [1], [2]).

- Human Preferences are not shown for Wam 2.1 experiments. Is the performance gain for Wan 2.1 and more powerful models not efficient enough? To address this concern, I recommend that you show the results of human preference experiments for Wan 2.1 1.3B/14B.

[1] Liu, et al. VideoDPO: Omni-Preference Alignment for Video Diffusion Generation. CVPR2025.
[2] Oshima, et al. Inference-Time Text-to-Video Alignment with Diffusion Latent Beam Search. NeurIPS2025.

**Questions:**

Please see the weakness for the major questions.

---

> ### Author Response · Authors · 2025-11-21
> **Response to Reviewer gKUH: iterative rounds (Part I)**
>
> >**Weakness1: The discussion on the number of iterative rounds is insufficient. It is unclear whether increasing the number of rounds continues to improve performance or saturates, and therefore, it is difficult to claim that an optimal recipe has been established.**
>
> We thank the reviewer for raising this point and apologize for not including experiments with a larger number of Dual-IPO rounds in the current draft. In fact, we did run **one additional round** beyond the setting reported in the paper. We found that VBench performance had essentially **saturated** (for example, on Wan-1.3B the overall VBench score only changed from **88.32 to 88.33**), while our internal reward continued to increase. This suggests that performance does **not** keep improving indefinitely with more iterations. Therefore, we do **not** claim to have discovered a globally *optimal* number of rounds; instead, our contribution is to provide a **practical iterative recipe** with early stopping based on VBench and human evaluation, which yields consistent gains over the baseline without inducing model collapse or severe overfitting to a particular reward.
>
> We did not include the extra-round results mainly for two reasons. First, after the sixth round the gain on VBench is extremely small (e.g., 88.32 → 88.33 for Wan-1.3B), making it difficult to draw strong conclusions: one could ask whether an additional round might eventually help, whether the current model has already reached its performance ceiling, or whether larger models would benefit more from further iterations. A more systematic study would also require additional evaluation metrics beyond VBench, which are currently scarce for video generation. Due to limited computational resources, we were not able to thoroughly explore these scaling questions in this work.
>
> Second, we observe that in later rounds our **internal reward model** continues to improve, but this no longer translates into measurable gains on VBench. On VBench’s 16 dimensions, most scores are already very high for our model; the remaining headroom mainly lies in **aesthetics** and **visual quality**. It is likely that by reweighting or redesigning our reward dimensions to more aggressively target these aspects, we could further increase the VBench score. However, this would amount to explicitly **tuning the reward to the benchmark**, which we view as less meaningful and closer to “benchmark hacking”. For this reason, we chose to stop training once VBench and human evaluation stopped improving, even though our internal reward kept rising.
>
> Overall, our method still **substantially alleviates reward hacking, overfitting, and model collapse** in RL-style video training, and it provides a practically useful recipe for **multi-round optimization** under reasonable stopping criteria. We fully agree that a truly complete “optimal recipe” will involve many engineering choices and further exploration of model size, number of rounds, and evaluation metrics. In the revised version, we will (i) report the additional-round experiment, and (ii) explicitly discuss the observed saturation behavior and the above limitations, so that our conclusions are properly contextualized.

---

> > ### Author Response · Authors · 2025-11-21
> > **Response to Reviewer gKUH: iterative rounds (Part II)**
> >
> > >**Weakness2: For the experiments on Wan, not all VBench metrics are reported.**
> >
> > We thank the reviewer for raising this concern. For Wan-1.3B, we only reported the total, quality, and semantic scores in Table 3 due to space limits. The full set of VBench metrics for Wan-1.3B (baseline and Dual-IPO) will be provided in the appendix. Regarding the trade-off between temporal consistency and dynamics, we have indeed paid attention to this issue in our experiments. We observe that the *Dynamic Degree* score is correlated with and affects many other metrics, and the **temporal-related scores remain relatively high** after training.
> >
> > Our analysis is that *Dynamic Degree* in VBench is computed based on an optical-flow algorithm, so camera motion or even slight camera shake can help maintain or increase this score. To better balance this, in our loss we incorporate both **real video–text pairs** and **model-generated samples that are rated high by the reward model**, and our reward explicitly emphasizes the motion dimension. This design reduces the negative impact that changes in dynamics might have on other dimensions. Overall, the remaining issues in dynamics and temporal coherence seem to stem more from limitations and ambiguities in the evaluation metrics and reward annotations than from the optimization procedure itself, and some of these aspects are inherently difficult to disentangle.
> >
> > | Model Name        |   Total   |  Quality  |  Semantic | Subject | Background | Temporal | Motion |  Dynamic  | Aesthetic |  Imaging  |
> > | ----------------- | :-------: | :-------: | :-------: | :-----: | :--------: | :------: | :----: | :-------: | :-------: | :-------: |
> > | Wan2.1-T2V-1.3B   |   84.26   |   85.30   |   80.09   |  96.34  |    97.29   |   99.49  |  97.44 |   85.56   |   62.43   |   66.51   |
> > | **Wan-1.3B-IPO3** | **86.29** | **86.39** | **85.88** |  97.12  |    96.08   |   99.77  |  97.90 |   92.44   |   64.61   |   66.35   |
> > | **Wan-1.3B-IPO5** | **88.31** | **87.84** | **90.21** |  97.63  |    97.61   |   99.75  |  97.94 | **95.37** | **67.81** | **68.35** |
> >
> > ---
> >
> > | Model Name        |   Object  | Multi-Obj |   Action  |   Color   |  Spatial  |   Scene   | Appearance |  Temporal |  Overall  |
> > | ----------------- | :-------: | :-------: | :-------: | :-------: | :-------: | :-------: | :--------: | :-------: | :-------: |
> > | Wan2.1-T2V-1.3B   |   94.3   |   79.74  |   98.2   |   89.58   |    76.46  |   52.73   |    21.28   |   25.32   |   27.07   |
> > | **Wan-1.3B-IPO3** | **94.72** | **92.95** | **97.05** |   85.01   |   94.38   |   69.77   |    23.51   |   26.24   |   25.31   |
> > | **Wan-1.3B-IPO5** | **97.06** | **95.01** | **96.84** | **92.43** | **98.37** | **82.54** |    23.59   | **27.54** | **26.91** |
> >
> > >**Weakness3: Human Preferences are not shown for Wam 2.1 experiments. Is the performance gain for Wan 2.1 and more powerful models not efficient enough? To address this concern, I recommend that you show the results of human preference experiments for Wan 2.1 1.3B**
> >
> > Thank you for pointing out the missing human preference evaluation for Wan2.1-T2V-1.3B/14B and for raising the question of efficiency on stronger backbones.
> >
> > In the current version, we mainly report human preference results on **CogVideoX-2B**, where we can clearly show the effect of different Dual-IPO iterations and the correlation with VBench. This choice was driven by the limited annotation budget and the very high cost of large-scale human evaluation for multiple backbones.
> >
> > At the same time, we fully agree that human judgments on **Wan2.1** are important. We are already running human evaluation for Wan2.1-T2V-1.3B, following exactly the same protocol as for CogVideoX-2B (pairwise A/B comparisons focusing on text–video consistency, motion, and overall quality). *Concretely, as described in the paper, for each prompt we generate one video from each model, and annotators rank the two videos to compute per-metric win rates. The experiments are in progress, and the results will be available shortly; we will include the preference rates and qualitative examples for both models in the revised version. However, due to resource limitations, we were unable to perform full reinforcement training on the 14B model, and therefore could not conduct evaluations. We hope you understand.
> >
> > The human evaluation results are as follows. On the Wan-1.3B model, our method (Dual-IPO) still has clear advantages, with substantial gains in semantic consistency, motion smoothness, and Faithfulness. Because wan's model is relatively better, the probability of a tie between the two is higher than that of cogvideo2B.:
> >
> > | Metric       | Wan-1.3B | Ties | Dual-IPO |
> > | ------------ | ------------ | ---- | -------- |
> > | Consistency  | 27        | 42.7 | 30.3     |
> > | Motion       | 23.4         | 29.1 | 47.5     |
> > | Faithfulness | 30.8         | 31.4 | 37.8    |
> >
> > Thanks again for your efforts and valuable insights, please let us know if you have any further questions.

---

> > > ### Comment · Reviewer_gKUH · 2025-11-26
> > >
> > > Thank you for the authors' detailed response.
> > > It would be beneficial to include these detailed explanations in your revised manuscript or supplementary materials.
> > > Most of my concerns have been addressed, so I will keep my initial positive score.

---

> > > > ### Author Response · Authors · 2025-11-26
> > > > **Official Comment by Authors**
> > > >
> > > > Thanks a lot for timely response. We have included these details in our revised version and will present more details in the supplementary material as suggested. We also plan to release all code, data, and models to ensure reproduction. Thanks again for your efforts and valuable insights that help us improve our paper.
> > > >
> > > > Best,
> > > >
> > > > Authors.

---

### Official Review · Reviewer_Qzdo · 2025-10-30

**Soundness:** 3
**Presentation:** 3
**Contribution:** 4
**Rating:** 6
**Confidence:** 4

**Summary:**

This paper presents Dual Iterative Preference Optimization (Dual-IPO), a novel post-training framework designed to enhance T2V generation models through iterative optimization of both the reward model and the video generation model. Unlike conventional alignment approaches such as DPO and KTO, which rely on fixed, human-annotated datasets, Dual-IPO introduces a bi-directional training loop. On the one hand, the reward model is iteratively refined via SRPO using CoT-guided reasoning, multi-path self-consistency voting, and a preference certainty estimator. The T2V model is updated based on dynamically evolving reward signals through Diffusion-DPO or Diffusion-KTO optimization. Experiments on CogVideoX and Wan show significant improvements in semantic alignment, motion smoothness, and aesthetic quality. The proposed framework achieves 81.33% human preference accuracy and strong VBench gains, establishing new SOTA results.

**Strengths:**

1.	The paper provides a motivation for addressing static reward limitations and distribution mismatch in preference alignment for generative video models.
2.	The proposed Dual-IPO is a novel optimization paradigm. The idea of jointly optimizing the reward model and generator in a feedback loop is interesting and novel. It addresses reward drift and distribution mismatch issues, which are common in previous methods like DPO or RLHF. It also reduce the requirements on large-scale human annotations.
3.	This paper proposes SRPO to train the reward model, which integrates CoT-guided reasoning, multi-path voting, PCE and SRPO loss. It supports both pairwise and pointwise supervision, which is flexible and unified for various alignment tasks.
4.	Dual-IPO achieves new state-of-of-art performance. Experiments on CogVideoX (2B/5B) and Wan (1.3B) show consistent gains (e.g., +3.0 on VBench), demonstrating robustness across architectures and parameter scales. The experiment validation is comprehensive and rigorous.

**Weaknesses:**

1.	The paper does not analyze the convergence of the dual-iterative process. There is no proof that the process will stably converge stably to an optimal point, rather than oscillating. The paper also does not clearly state when the iterative process should stop.
2.	Lack of ablation study. For example, the paper does not contain the complete ablation study of the three key parts of SRPO (CoT, self-consistency and PCE).
3.	The method is complicated and inefficient, which is pointed out by the authors. It requires 128 GPUs and approximately two weeks per iteration, making it extremely resource-intensive. The paper does not analyze the efficiency of each component, or explore the strategy to reduce computational cost.
4.	It would be better and clearer to illustrate SRPO and its components with graphs.
5.	Some typos (eg. in Figure 1 (c)).

**Questions:**

1.	What is the convergence criteria of the dual-iterative preference optimization? Will the excessive iterations cause overfitting? Is there a risk of reward hacking or mode collapse in later iterations?
2.	What is the bottleneck that restricts the computational efficiency? Have you explored any strategies to reduce computational requirements? How does the performance scale with reduced computational budgets?
3.	Could you provide the ablation study of CoT, self-consistency and PCE to prove their effectiveness individually? And what if replace SRPO with static DPO?
4.	Could Dual-IPO be generalized beyond T2V? The design seems extensible.

---

> ### Author Response · Authors · 2025-11-21
> **Response to Reviewer Qzdo: convergence criteria and computational efficiency (Part I)**
>
> >**Weaknesses1: The paper does not analyze the convergence of the dual-iterative process. There is no proof that the process will stably converge stably to an optimal point, rather than oscillating. The paper also does not clearly state when the iterative process should stop.**
>
> We thank the reviewer for this very helpful suggestion and for highlighting an important aspect of our method. We fully agree that both (i) the theoretical underpinnings of the dual-iterative process and (ii) the stopping criterion and convergence behavior of our iterations should be stated more clearly. On the first point, **our design is indeed supported by existing theoretical and empirical work**; however, due to the scale of our experiments and space constraints, we did not fully elaborate on these connections in the original submission. We will briefly summarize these foundations below and clarify them in the revised manuscript. On the second point, in practice we stop the iterative process when additional Dual-IPO iterations bring only very limited improvements in VBench. We believe this saturation is partly due to our current reward design and the limited coverage of existing video-generation metrics, as well as potential capacity limits of smaller models. Because of resource constraints we were unable to run very long iteration sweeps, but we will include curves from extended training runs to illustrate the observed convergence/saturation behavior in the revision.
>
> First, our design is motivated by prior work on iterative RLHF-style training, which shows (i) the necessity of updating the reward model during training and (ii) the effectiveness of using data labeled by the current model to iteratively improve the reward model’s accuracy and alignment [1,2]. In these pipelines, the policy and reward model are updated in alternating stages rather than via a single offline pass. This directly inspires our SRPO component, where the critic is refined with self-labeled preference data filtered by a certainty estimator, thereby reducing label noise as the policy improves. These theoretical and empirical insights guided the design of our dual-iterative scheme, though they were not sufficiently emphasized in the original text; we will make this connection explicit in the revised version.
>
> Second, the effectiveness of iterative DPO updates is supported by the standard RL objective used in DPO-like analyses [3,4]. The optimal aligned policy can be written as
>
> $$
> \pi^*(x_0 \mid c) \propto \pi_0(x_0 \mid c)\,\exp\left(\frac{1}{\beta} r(x_0, c)\right).
> $$
>
> where $(\pi_0)$ is the reference policy and $(r)$ is the reward. Online RL methods such as PPO and GRPO can be viewed as stochastic optimization procedures that $(approximately)$ move $(\pi_0)$ toward $(\pi^\*)$ with *online* data collection, and they often outperform a single offline DPO step in practice. This motivates our iterative optimization scheme: by repeatedly updating $(\pi_0)$ with DPO/KTO under an improved reward model, we expect $(\pi_0)$ to gradually approach $(\pi^*)$, leading to more stable and better performance than a one-shot optimization.
>
> Regarding stability, we agree that mild oscillations are inevitable in reinforcement-learning-style training, and we do not claim that the loss curves are perfectly monotone. In our paper, “stability” specifically refers to controlled behavior of the system. Concretely, in each Dual-IPO iteration we monitor VBench scores and training losses; when we observe signs of degradation or instability (e.g., VBench scores drop, or losses behave pathologically), we trigger a new Dual-IPO iteration and refine the critic. In the reported experiments, the metrics consistently improve and then saturate, rather than oscillating wildly or diverging. We will make this empirical notion of stability (and the corresponding plots, including later-iteration curves) more explicit in the revised manuscript.
>
> [1] Llama: Open and efficient foundation language models
>
> [2] Llama 2: Open foundation and fine-tuned chat models
>
> [3] Mathematical analysis of machine learning algorithms.
>
> [4] Iterative Reasoning Preference Optimization

---

> > ### Author Response · Authors · 2025-11-21
> > **Response to Reviewer Qzdo: convergence criteria and computational efficiency (Part II)**
> >
> > >**Weaknesses2: Lack of ablation study. For example, the paper does not contain the complete ablation study of the three key parts of SRPO (CoT, self-consistency and PCE).**
> >
> > We thank the reviewer for pointing out the importance of ablations. In fact, Table 6 (As shown in the table below) already presents ablations of the main SRPO components, including the SRPO loss and the preference-certainty estimator (PCE), as well as a comparison between full SRPO and a static DPO baseline. These experiments demonstrate that (i) SRPO consistently outperforms static DPO, and (ii) both the SRPO loss and PCE contribute to the final performance.
> >
> > | Setting     | Critic Update | SRPO loss | PCE | VBench Score |
> > |------------|---------------|-----------|-----|--------------|
> > | Baseline   | ✗             | —         | —   | 82.74        |
> > | No update  | ✗             | —         | —   | 82.33        |
> > | SRPO (full)| ✓             | ✓         | ✓   | **82.91**    |
> > | SRPO loss  | ✓             | ✗         | ✓   | 82.83        |
> > | PCE        | ✓             | ✓         | ✗   | 82.69        |
> >
> >
> > Regarding CoT and self-consistency, we would like to clarify their role. CoT-guided reasoning is the basis of our pseudo-labeling pipeline: without CoT there is no multi-path reasoning and thus no self-consistency voting. In this case, the method essentially degenerates to a static DPO-style scheme with single-path labels, i.e., the static DPO baseline already reported in Table 6. Therefore, the effect of removing CoT and self-consistency is indirectly reflected by the comparison between SRPO and static DPO.
> >
> > The usefulness of CoT and self-consistency for improving label quality has been well established in prior work [5,6]. For this reason, we focused our ablations on the parts that are novel to SRPO (the SRPO loss and PCE) and on the SRPO vs. static DPO comparison, and we will make this connection to Table 6 more explicit in the revised version.
> >
> > [5] Let's Verify Step by Step
> >
> > [6] Self-Consistency Improves Chain of Thought Reasoning in Language Models
> >
> > >**Weaknesses3: The method is complicated and inefficient, which is pointed out by the authors. It requires 128 GPUs and approximately two weeks per iteration, making it extremely resource-intensive. The paper does not analyze the efficiency of each component, or explore the strategy to reduce computational cost.**
> >
> > We thank the reviewer for this concern and fully agree that computational efficiency and scalability are important practical issues. Our current experiments indeed use a large configuration to push performance, and we explicitly list this as a limitation. At the same time, we would like to clarify (i) where the computational bottlenecks actually come from, (ii) in what sense Dual-IPO is designed to be small-lab friendly, and (iii) what practical trade-offs and cost-reduction strategies are available.
> >
> > In summary, we highlight three points:
> >
> > - The dominant cost in our pipeline comes from video generation and evaluation (including VBench), which any strong preference-based alignment method on large T2V models must pay; the extra overhead of Dual-IPO itself (critic updates, SRPO) is relatively small.
> >
> > - Dual-IPO is explicitly designed to reduce human annotation cost compared to standard RLHF-style pipelines, while keeping the computational overhead modest and largely reusable across models.
> >
> > - The large configuration reported in the paper is a best-practice setting for pushing SOTA, not a requirement of the algorithm. In practice, smaller generators and critics, fewer samples per iteration, and fewer iterations already yield substantial gains (e.g., a 2B CogVideoX surpassing the 5B baseline after three iterations).
> >
> > Compared with previous approaches that rely on tens of thousands to hundreds of thousands of manually labeled preference pairs and repeated reward-model training, our main focus is to reduce human annotation cost while still allowing the reward model to be updated in the middle of training. In many existing pipelines, the total human labeling time can easily exceed the training time, and the reward model is difficult to refresh once deployed. In contrast, SRPO uses a small amount of human labels together with automatic pseudo-labeling and self-refinement, which makes our approach more efficient and practical from the perspective of annotation and maintenance cost.
> >
> > The configuration we report in the paper is a “best-practice” setting, not the only possible one. In real applications there are multiple trade-offs that can significantly reduce compute: a smaller reward model can already achieve comparable gains, and a smaller T2V model benefits strongly from Dual-IPO (e.g., CogVideoX-2B surpassing the 5B baseline). Moreover, most of the wall-clock time is spent on video generation itself, which is a known bottleneck for RL-based video training (also noted in DanceGRPO), rather than on our optimization algorithm.

---

> > > ### Author Response · Authors · 2025-11-21
> > > **Response to Reviewer Qzdo: convergence criteria and computational efficiency (Part III)**
> > >
> > > >**Weaknesses4: It would be better and clearer to illustrate SRPO and its components with graphs.**
> > >
> > > We thank the reviewer for this helpful suggestion. We agree that SRPO and its components can be made much clearer with graphical illustrations, and we will revise the paper accordingly.
> > >
> > > In the revised version, we have added a detailed flowchart of the SRPO algorithm, including PCE filtering and human validation. The anonymous figure is available at:
> > > [https://anonymous.4open.science/r/ICLR2026_Dual_IPO-5262/SRPO_figure.png](https://anonymous.4open.science/r/ICLR2026_Dual_IPO-5262/SRPO_figure.png)
> > > This diagram visualizes the full SRPO iteration: sampling videos from the current T2V model, applying PCE to filter high-certainty pairs, generating pseudo-labels, updating the critic, and then validating the updated critic.
> > >
> > > In our current implementation, each SRPO iteration is accompanied by human validation: after every update, we evaluate the critic on a human-labeled preference test set, where annotators rank outputs from the current T2V model. On this set, we compare our critic with open-source reward models and report the human agreement in Table 1, and in Table 6 we track both this agreement and downstream VBench metrics before and after SRPO updates. In practice, we only accept SRPO updates when the agreement with human labels stays above a fixed threshold (75% in our experiments), and we also use the VBench metrics to decide whether the pseudo-label training is beneficial or whether the critic should be retrained.
> > >
> > > We will integrate the new SRPO flowchart into the paper and expand the corresponding section to explicitly walk through this iteration process, so that the role of each component (SRPO loss, PCE, and human validation) is clear from both the text and the figures.
> > >
> > >
> > > >**Weaknesses5: Some typos (eg. in Figure 1 (c)).**
> > >
> > > We thank the reviewer for pointing this out. We will carefully proofread the paper and correct all typos and minor inconsistencies, including those in Figure 1(c), in the revised version.

---

> > > > ### Author Response · Authors · 2025-11-21
> > > > **Response to Reviewer Qzdo: convergence criteria and computational efficiency (Part IV)**
> > > >
> > > > >**Question 1: What is the convergence criteria of the dual-iterative preference optimization? Will the excessive iterations cause overfitting? Is there a risk of reward hacking or mode collapse in later iterations?**
> > > >
> > > > We thank the reviewer for this important question. Our dual-iterative preference optimization is explicitly designed to mitigate reward hacking, overfitting, and mode collapse: we use a practical stopping criterion based on external metrics, we refresh the reward model when we detect a mismatch between its scores and external metrics, and when the T2V model’s performance starts to degrade we reset the reference policy to the best previous checkpoint before continuing optimization. This combination empirically prevents the model from drifting toward degenerate solutions even if the internal reward keeps increasing.
> > > >
> > > > Concretely, after a certain number of training steps in each iteration, we monitor three signals—(1) VBench scores on held-out prompts, (2) validation losses, and (3) human side-by-side preference against the pre-iteration model—and we stop or roll back the current iteration if these signals saturate or worsen. In exploratory runs, when we pushed to later iterations, VBench and human-perceived improvements became very small while our own reward model’s score continued to rise; this suggests that at the current model scale further progress likely requires adjusting the evaluation dimensions of our reward model and retraining the critic, and that the generator may already be close to its capacity limit. Due to resource constraints, we did not conduct a more exhaustive analysis in this very late-iteration regime.
> > > >
> > > > >**Question 2:What is the bottleneck that restricts the computational efficiency? Have you explored any strategies to reduce computational requirements? How does the performance scale with reduced computational budgets?**
> > > >
> > > > We thank the reviewer for this question. In our full training pipeline, the dominant bottleneck is the *video data generation and evaluation*, rather than the SRPO/DPO optimization itself. For each Dual-IPO iteration, we need to (i) generate a large number of videos for preference data and (ii) repeatedly run VBench, which itself requires generating many videos and running a non-trivial scoring pipeline. These costs are largely unavoidable if we want diverse preference data and reliable evaluation.
> > > >
> > > > For DPO-style optimization, we have explicitly explored how the number of samples per prompt affects both performance and compute. On the same prompt, we tried generating 2, 4, 8, and 12 videos and forming preference pairs from them. Under our current (relatively small) model scale, 8 samples per prompt gives the best trade-off between accuracy and computational cost: using only 2 samples significantly reduces diversity and makes preference pairs less informative, leading to only very small gains over the baseline, while going beyond 8 samples brings only marginal improvements despite much higher sampling cost. Therefore, we adopt 8 samples per prompt as our default choice in the main experiments. On the evaluation side, VBench similarly requires multiple samples per prompt and multiple runs during training, which further contributes to the wall-clock cost.
> > > >
> > > >
> > > > >**Question 3:Could you provide the ablation study of CoT, self-consistency and PCE to prove their effectiveness individually? And what if replace SRPO with static DPO?**
> > > >
> > > > As explained in our response to Weakness 2, Table 6, we already ablate the main SRPO components: removing PCE and replacing the SRPO loss with a standard DPO loss both degrade performance, and full SRPO clearly outperforms the static DPO baseline, demonstrating the effectiveness of SRPO and PCE individually. For CoT and self-consistency, these are the basis of our pseudo-labeling pipeline: without CoT there is no multi-path reasoning and thus no self-consistency voting, so the method essentially degenerates to a static DPO-style scheme with single-path labels, i.e., the static DPO baseline already reported in Table 6. Therefore, the effect of CoT and self-consistency is indirectly reflected in the SRPO vs. static DPO comparison.
> > > >
> > > > >**Question 4: Could Dual-IPO be generalized beyond T2V? The design seems extensible.**
> > > >
> > > > Dual-IPO is designed to be extensible beyond T2V. The core ingredients, including iterative DPO-style optimization, self-consistency based pseudo labeling, and SRPO for reward refinement, are modality agnostic and only require a conditional generative model and a reward model. We chose to instantiate and evaluate Dual-IPO on T2V because video preference data are particularly hard and costly to annotate, existing reward models have limited coverage and generalization, and many recent systems (e.g., Wan, HunyuanVideo) are not well supported by current open-source preference pipelines. However, the same framework can in principle be applied to text-to-image and language models.

---

> > > > > ### Author Response · Authors · 2025-12-01
> > > > > **Gentle Reminder as Discussion Phase Closing Soon**
> > > > >
> > > > > Dear Reviewer Qzdo,
> > > > >
> > > > > We sincerely appreciate your reviews and feedback in the earlier stage. Your comments are helpful for us and the work.
> > > > >
> > > > > We have submitted our rebuttal and would be delighted to continue the discussion with you. Besides the revision and response, we also plan to release all code, data, and models to ensure reproduction. Thanks again for your efforts and valuable insights that help us improve our paper. As the discussion phase deadline is approaching, we just wanted to kindly check whether our responses have addressed your concerns.
> > > > >
> > > > > Your perspective is extremely important to us, and we would be truly grateful for any further thoughts you might share before the discussion closes. Thank you again for your time and contributions.
> > > > >
> > > > > Best regards,
> > > > >
> > > > > Authors.

---

### Official Review · Reviewer_bijg · 2025-10-31

**Soundness:** 3
**Presentation:** 3
**Contribution:** 3
**Rating:** 6
**Confidence:** 4

**Summary:**

This paper proposes Dual-Iterative Preference Optimization (Dual-IPO), a post-training framework that iteratively refines both a reward model and a text-to-video (T2V) generator to better align video synthesis with human preferences.
The method combines:

- Self-Refined Preference Optimization (SRPO) — leveraging Chain-of-Thought (CoT) guided reasoning, self-consistency voting, and a Preference Certainty Estimator (PCE) for robust pseudo-labeling.

- Dual iterative optimization — alternately improving the critic (reward) model and the video generation model via both Diffusion-DPO and Diffusion-KTO objectives.
Experiments on CogVideoX, Wan, and VBench benchmarks show consistent gains in motion smoothness, aesthetic quality, and prompt consistency, with a 2B model surpassing a 5B baseline.

**Strengths:**

- Dual optimization design – The interplay between a self-refined reward model and iterative generator updates is conceptually elegant and empirically validated.

- Methodological soundness – The CoT-guided pseudo-labeling and PCE-weighted DPO/KTO training are carefully formulated and ablated.

- Strong experimental validation – Includes results across model sizes, architectures, and both automatic and human evaluations, showing consistent improvement.

- Clarity and completeness – Writing, figures, and methodology are clear; the paper includes reproducibility and ethics statements, and provides rich supplementary material.

**Weaknesses:**

- High computational cost – Each optimization round involves dual training of large models (VILA-40B, CogVideo-5B), making it unclear how scalable or practical Dual-IPO is for typical research labs.

- Dependence on synthetic preference labels – Despite PCE filtering, the pseudo-label quality may still drift; more human validation or robustness analysis would strengthen the claims.

**Questions:**

- How sensitive is Dual-IPO to the initial human preference dataset size and quality? Would small or noisy seeds destabilize the dual-loop training?

- Do you observe any reward overfitting or “reward hacking” effects in later iterations? If so, how does SRPO mitigate them?

- Could the authors provide a more quantitative analysis of the trade-off between computational cost and performance gains? Specifically, each iterative round seems to require large-scale training with multiple pseudo-labeling and re-optimization stages. How does performance improvement scale with training cost (e.g., GPU hours vs. VBench gain)? Is there an observed saturation point after which additional iterations bring diminishing returns?

---

> ### Author Response · Authors · 2025-11-21
> **Response to Reviewer bijg: Cost–Performance Trade-off and Saturation Behavior (Part I)**
>
> >**Weaknesses 1: High computational cost – Each optimization round involves dual training of large models (VILA-40B, CogVideo-5B), making it unclear how scalable or practical Dual-IPO is for typical research labs.**
>
> We thank the reviewer for raising this concern. First, in our current system the dominant practical bottleneck comes from video sampling speed and VBench evaluation time, rather than from the Dual-IPO algorithm itself. Second, Dual-IPO is explicitly designed to be small-lab friendly by reaching strong (often SOTA-level) performance with small generators and a lightweight, reusable critic. We elaborate on these two points below.
>
> A key motivation of Dual-IPO is to **avoid continuous human annotation**, which is usually the main bottleneck in RLHF-style pipelines. In many RLHF pipelines (e.g., for LLaMA-style models[1][2]), the reward model is repeatedly updated on fresh model outputs using **large-scale human preference datasets** and each new RL round requires collecting additional human labels, which is extremely time- and labor-intensive and hard to reproduce for typical labs. In contrast, our reward model updates are fully automated once a small seed set is prepared: one SRPO-based critic update in our implementation takes roughly one day on 8×A100 GPUs, without any extra human annotation in later rounds. This is exactly the starting point of our work: to reduce the labeling demand for the model being optimized and to accelerate the iterative optimization of the reward model. Overall, compared to standard RLHF-style pipelines, Dual-IPO is relatively lightweight and friendly to small research labs, as it avoids repeated human annotation and keeps the main compute in one-time synthetic data generation and evaluation.
>
> Importantly, Dual-IPO is designed to reach strong performance starting from small generators. Our main gains come from aligning a 2B CogVideoX model, which achieves SOTA-level VBench performance and surpasses the 5B baseline after only three Dual-IPO iterations. This is a scale much closer to what typical research labs can afford.
> On the critic side, we only fine-tune a pre-trained VILA (13B/40B) reward model for 5 epochs*on 16 video frames with a small batch size on 32 GPUs, and then apply SRPO to generate pseudo-labels for further refinement. This is a modest post-training stage compared to full T2V pretraining and can be reused across generators.so labs with limited resources can simply run fewer iterations or use smaller critics to trade cost for performance in a controlled way.
>
> [1] Llama: Open and efficient foundation language models
>
> [2] Llama 2: Open foundation and fine-tuned chat models

---

> > ### Author Response · Authors · 2025-11-21
> > **Response to Reviewer bijg: Cost–Performance Trade-off and Saturation Behavior (Part II)**
> >
> > >**Weaknesses 2: Dependence on synthetic preference labels – Despite PCE filtering, the pseudo-label quality may still drift; more human validation or robustness analysis would strengthen the claims.**
> >
> > We appreciate the reviewer’s insightful comment on the dependence on synthetic preference labels, and we apologize that our original presentation did not make this logic sufficiently clear. In our current implementation, each SRPO iteration is accompanied by human validation: after every update, we evaluate the critic on a human-labeled preference test set, where annotators rank outputs from the current T2V model.
> > We have added a more detailed flowchart of the SRPO algorithm, including PCE filtering and manual verification. The anonymous link is below:https://anonymous.4open.science/r/ICLR2026_Dual_IPO-5262/SRPO_figure.png
> > On this set, we compare our critic with open-source reward models and report the human agreement in Table 1, and in Table 6 we track both this agreement and downstream VBench metrics before and after SRPO updates. In practice, we only accept SRPO updates when the agreement with human labels stays above a fixed threshold (75% in our experiments), and we also use the VBench metrics to decide whether the pseudo-label training is beneficial or whether the critic should be retrained. We will additionally add a more detailed illustration of the SRPO iteration process in the revised version to make this procedure explicit in the pipeline.
> >
> > >**Questions1:How sensitive is Dual-IPO to the initial human preference dataset size and quality? Would small or noisy seeds destabilize the dual-loop training?**
> >
> > We thank the reviewer for the question and acknowledge that we did not perform a strict ablation on injected noisy seeds. Our goal in this work is not a fully generic large-scale reward model, but a model-specific critic trained on a small, high-quality preference set, so we focus on annotator training, guideline refinement, category balancing, and filtering rather than sheer data volume. We expect more clean data to further help, and will add analysis and discussion on the impact of seed size/quality and filtering strategies in the revision.
> >
> > >**Questions 2:Do you observe any reward overfitting or “reward hacking” effects in later iterations? If so, how does SRPO mitigate them?**
> >
> > We thank the reviewer for the insightful question, and we apologize for not explaining this clearly in the original submission. Yes, we do observe “reward hacking” in our experiments. Around the 3rd–4th SRPO iteration, the internal reward score kept increasing while VBench dropped, indicating that the generator was overfitting to the existing reward model. After updating the reward model with SRPO (using human-validated pseudo labels), this mismatch was significantly alleviated and VBench recovered, showing that iteratively adapting the reward model is an effective way to mitigate such reward hacking.
> > In later iterations (beyond ~6), we observe that VBench improvements become very marginal, even though the reward given by our critic continues to increase. We conjecture two reasons for this: (i) the current small T2V model may already be close to its performance ceiling under VBench, and (ii) the evaluation dimensions emphasized by our reward model may need to be adjusted, for example by placing more weight on aesthetics and visual quality, where the VBench scores are relatively lower than other dimensions.

---

> > > ### Author Response · Authors · 2025-11-21
> > > **Response to Reviewer bijg: Cost–Performance Trade-off and Saturation Behavior (Part III)**
> > >
> > > >**Questions 3: Could the authors provide a more quantitative analysis of the trade-off between computational cost and performance gains?**
> > >
> > > We thank the reviewer for this detailed question, and we apologize that the trade-off was not explained clearly in the original submission. In terms of sampling cost, we already study the effect of generating different numbers of videos per prompt when constructing preference pairs. Concretely, for each prompt we generate multiple candidates and keep only the best and worst videos (discarding the middle ones). We experiment with generating 2, 4, 8, and 10 candidates per prompt, and find that 8 candidates provide the best balance between performance and wall-clock cost: more candidates generally improve robustness, but the marginal gains beyond 8 are small compared to the additional sampling time. This reflects the classic utilization–efficiency trade-off in RL-style methods, where more samples help but quickly become prohibitively expensive.
> > >
> > > Regarding the number of Dual-IPO rounds and total training compute, as discussed in our response to Question 2, the current small model already appears close to its performance limit under VBench, and the overall trade-off depends not only on compute but also on the design of the reward model and evaluation metrics. In particular, our VBench scores suggest that aesthetics and visual quality remain relatively weaker dimensions, indicating that the reward model’s emphasis may also need to be adjusted, not just the amount of compute. For these reasons we did not further over-optimize the number of rounds in this work, although in principle allocating more compute to sampling and better-aligned reward dimensions should yield additional gains. We will clarify this discussion in the revised version.

---

> > > > ### Comment · Reviewer_bijg · 2025-11-28
> > > >
> > > > Thank you to the authors's thorough response. Most of my concerns have been satisfactorily addressed, so I will maintain my original positive rating.

---

> > > > > ### Author Response · Authors · 2025-12-01
> > > > > **Official Comment by Authors**
> > > > >
> > > > > Thank you very much for your constructive suggestions and we will include these detailes in our revised version. We will also release all code, data, and models to ensure reproduction. Feel free to let us know if you have any further questions.
> > > > >
> > > > > Best,
> > > > >
> > > > > Authors.

---

### Official Review · Reviewer_pm1R · 2025-11-01

**Soundness:** 3
**Presentation:** 3
**Contribution:** 3
**Rating:** 4
**Confidence:** 3

**Summary:**

This paper proposes Dual-IPO (Dual-Iterative Preference Optimization), a novel post-training framework for text-to-video (T2V) generation that enhances both synthesis quality and alignment with human preferences. The method involves a dual iterative paradigm that sequentially optimizes a reward model and a video generation model. The reward sign is refined through Chain-of-Thought (CoT) guided reasoning, voting-based self-consistency, and a preference certainty estimator to ensure reliable feedback. Using these improved reward signals, the video generation model is iteratively updated to enhance semantic consistency, motion smoothness, and aesthetic quality. Through multiple optimization rounds, both models are progressively improved without the need for extensive manual preference annotations. Experimental results show that Dual-IPO substantially improves video generation quality and allows smaller models to outperform larger baselines.

**Strengths:**

1) The authors introduce a comprehensive and well-motivated dual-iterative framework that jointly optimizes the reward and generation models, addressing limitations of static preference alignment methods.

2) The proposed approach demonstrates strong data efficiency, requiring only a small amount of human-annotated preference data to initiate the self-refinement process.

3) The paper effectively captures the evolving nature of human preferences, emphasizing that fixed offline datasets may lead to overfitting and reduced generalization.

4) Extensive experiments across multiple model architectures and scales (e.g., CogVideoX and Wan) are provided, supported by quantitative metrics (VBench) and qualitative analysis.

**Weaknesses:**

1) The construction of textual prompts used for generating training data is under-specified. The authors mention the use of structured elements (subjects, attributes, spatial relations, and actions), but the paper would benefit from more details regarding the **size, diversity, and balance of the prompt pool**. For instance, how many combinations were used per category, and how does this affect diversity and representativeness?

2) The construction of textual prompts may inadvertently introduce **bias** into the generated training data. Since prompts reflect the authors’ design choices, such as the selection of subjects, attributes, and actions. They could encode cultural, demographic, or semantic biases present in the prompt corpus. For example, certain object, action combinations might be overrepresented, or specific demographic groups may be underrepresented if not explicitly balanced. Such biases could propagate through the downstream model, affecting its generalization and fairness. The paper would benefit from a discussion of how prompt templates were curated, whether any bias detection or mitigation strategies (e.g., prompt balancing, debiasing filters, or controlled generation) were applied, and how residual biases were evaluated or monitored.

**Questions:**

My main concern regarding this paper is the diversity of the user-specified training data construction and the potential bias led by this procedure. If the authors could address my concern effectively. I'd be happy to raise my score.

---

> ### Author Response · Authors · 2025-11-21
> **Response to Reviewer pm1R: Textual Prompt Construction and Bias Mitigation (part I)**
>
> >**Weaknesses 1:The construction of textual prompts used for generating training data is under-specified. The authors mention the use of structured elements (subjects, attributes, spatial relations, and actions), but the paper would benefit from more details regarding the size, diversity, and balance of the prompt pool. For instance, how many combinations were used per category, and how does this affect diversity and representativeness?**
>
> We thank the reviewer for the valuable suggestion. You are correct that the construction of textual prompts plays a crucial role in our framework. We have added Figure 1 in the appendix, which reports the distributions of the sub-categories for both subjects and actions.
> We provide anonymous links to generate prompts as follows: https://anonymous.4open.science/r/ICLR2026_Dual_IPO-5262/create_prompt.py
>
> Below we describe our prompt construction procedure in more detail.
>
> **Size.**
> Our prompts are generated from structured templates with random instantiation rather than a fixed list, so the pool is effectively unbounded. For each experiment, we sample many prompts on the fly and can adjust the sampling distribution to emphasize different categories as needed.
>
> **Diversity.**
> The templates factor subjects, attributes, spatial relations, and actions, populated from rich vocabularies. For humans, we vary age, occupation, gender, and forms of address; for animals, we cover a large number of common species plus alternative names; for other categories (objects, food, vehicles, etc.), we span everyday and visually distinctive concepts. These ingredients are combined into single- and multi-subject scenes with explicit spatial relations, and a large language model (LLM) is used to expand and filter prompts. Clearly unrealistic combinations are not discarded but mapped to stylized domains such as “cartoon”, “dreamlike”, or “sci-fi” so that they remain meaningful.
>
> **Balance.**
> Because the pool is large and diverse, we enforce balance at sampling time: during training, we control the proportion of different categories (humans, animals, objects, single vs. multiple targets, easy vs. harder actions) so that each epoch sees a roughly balanced mixture of content types rather than being dominated by a single class.
>
> For **how many combinations per category**, We do not fix a list of prompts per category: in each epoch we sample from structured pools under a target type distribution, so combinations per category are very large and rarely repeat, and our templates emphasize diverse multi-object scenes and motions.
>
> A more detailed description is as follows.
>
> 1. **Structured subject space.** We organize subjects into two main groups, humans and animals. Humans include multiple age and gender categories and many professions, combined with attributes such as body shape and hair properties. Animals are divided into land, flying, and marine species, and we enumerate a large set of concrete types. Subjects can appear as single or multiple targets; for multi-subject prompts, we additionally sample spatial relations (left/right/above/below, etc.) to create structured multi-object configurations. An LLM filters implausible combinations; a small fraction of intentionally unrealistic ones are kept but explicitly marked as abstract scenarios and expanded into longer descriptions.
>
> 2. **Action taxonomy.** To better cover temporal behaviors, we categorize actions into simple, medium, and difficult (e.g., walking/running vs. jumping/crawling vs. complex sports actions). Actions are combined with subjects and filtered by the LLM; non-realistic subject–action pairs are restricted to a small fraction of the pool and treated as abstract scenes with explicitly stylized captions.
>
> 3. **Additional object categories.** Beyond humans and animals, we include other categories such as vehicles, food, and common objects. These are combined with human/animal subjects, enriched with attributes, and processed by the same LLM-based expansion and filtering pipeline.
>
> 4. **Scene contexts and sampling schedule.** We define many scene types, grouped into realistic scenes (e.g., snowfield, forest, farmland, park) and abstract scenes (e.g., a car driving inside a coffee cup). During sampling, we control the probability over prompts so that it gradually decreases from easy to hard actions, realistic to abstract scenes, single- to multi-subject compositions, and everyday to highly surreal situations, keeping the dataset dominated by realistic scenarios while still providing a diverse long tail of challenging combinations.
>
> Overall, this structured, programmatic pipeline makes the prompt space both controllable and effectively unbounded, and ensures that the resulting prompt pool is large, diverse, and reasonably balanced across categories. We will release the main prompt templates and vocabularies used in our construction to facilitate reproducibility.

---

> ### Author Response · Authors · 2025-11-21
> **Response to Reviewer pm1R: Textual Prompt Construction and Bias Mitigation  (part II)**
>
> >**Weakness 2: The construction of textual prompts may inadvertently introduce bias into the generated training data. Since prompts reflect the authors’ design choices, such as the selection of subjects, attributes, and actions.**
>
> We appreciate this concern and note that these potential biases were already taken into account when designing our prompt construction pipeline: our prompt pool is constructed with broad coverage and is processed through multiple LLM-based filtering stages. In each training run, we further fix a target distribution over high-level categories and sample from the global prompt pool accordingly; because subjects and attributes are drawn from broad, approximately uniform vocabularies, individual prompts almost never repeat and no single category is systematically underrepresented, while later stages place more emphasis on challenging compositions involving multiple subjects, attributes, spatial relations, actions, and abstract scenes.
>
> We agree that textual prompts can be a source of cultural and demographic bias, and we explicitly considered this when designing our prompt pipeline.
>
> First, we use a controllable, enumerated construction of subjects and attributes. For humans, we explicitly list variants over age, gender-neutral and gendered roles (e.g., child, adult, elderly; teacher, student, firefighter, musician, etc.), as well as physical attributes such as height, body shape, and hair style/color. For animals, we similarly enumerate different surface forms and a large number of concrete species. These subject tokens are sampled approximately uniformly from their lists, and the high-level prompt categories (e.g., humans vs. animals vs. objects, single vs. multiple subjects, easy vs. harder actions) are controlled by explicit sampling weights, so that no single demographic group or object type is systematically underrepresented. Actions are also stratified by difficulty, and their sampling probabilities are chosen according to the capability of the base model (e.g., placing more mass on simple actions for smaller backbones), which avoids over-emphasizing rare complex motions.
>
> Second, we use the LLM in a restricted and filtered manner. The core discrete choices—subject, attribute, action, scene—are drawn randomly from the structured pools; the LLM is only used to expand these tuples into natural-language descriptions, not to decide which demographic or object appears. We further apply an automatic filtering step (implemented via the LLM) that removes prompt–caption pairs that are nonsensical, unsafe, or clearly discriminatory or offensive, and we perform light manual spot checks over random batches to correct or discard problematic patterns. During training, prompts are sampled from the global pool under the above category-wise sampling weights, and we monitor the empirical distribution over subject and action categories to keep it roughly balanced.
>
> Overall, this design makes the prompt space both controllable and auditable: it allows us to balance demographic and semantic coverage more effectively than using scraped “real” prompts, while still supporting a large, diverse set of combinations. In the revision, we will add a short subsection describing these curation and filtering steps, together with summary statistics of the prompt distributions, to make our bias mitigation strategy more explicit. We also acknowledge that residual biases may remain, and view a more systematic fairness analysis as an important direction for future work.
>
> Thanks again for your efforts and valuable insights, please let us know if you have any further questions.

---

> > ### Comment · Reviewer_pm1R · 2025-11-23
> > **Response to the author rebuttal**
> >
> > Thanks for the detailed response from the authors. It would be beneficial to include these detailed explanations in your revised manuscript or supplementary materials. Most of my concerns have been addressed and I am happy to raise my initial score to 6.

---

> > > ### Author Response · Authors · 2025-11-24
> > > **Official Comment by Authors**
> > >
> > > Thank you very much for your constructive suggestions and we will include these detailes in our revised version. We will also release all code, data, and models to ensure reproduction. Feel free to let us know if you have any further questions.
> > >
> > > Best,
> > >
> > > Authors.

---

### Author Response · Authors · 2025-12-01
**Brief Summarization of Rebuttal and Revisions (Part2/2)**

**4. Convergence analysis of our framework**

Reviewer `Qzdo` noted the importance of convergence analysis of our method. Accordingly, we discussed the convergence and stability of our method as detailed in the rebuttal below.

**5.Ablation study on the three key parts of SRPO (CoT, self-consistency and PCE).**

Reviewer `Qzdo` mentioned the ablation study of three key parts of SRPO. In Table 6, we present ablations of the main SRPO components, including the SRPO loss and the preference-certainty estimator (PCE), as well as a comparison between full SRPO and a static DPO baseline. These experiments demonstrate that (i) SRPO consistently outperforms static DPO, and (ii) both the SRPO loss and PCE contribute to the final performance.

**6.Discussion on the number of iteative round**

Reviewer `gKUH` pointed out that the importance of discussion on the number of iterative round. Actually, we did run one additional round beyond the setting reported in the paper. We found that VBench performance had essentially saturated (for example, on Wan-1.3B the overall VBench score only changed from 88.32 to 88.33), while our internal reward continued to increase. This suggests that performance does not keep improving indefinitely with more iterations. Therefore, we do not claim to have discovered a globally optimal number of rounds; instead, our contribution is to provide a practical iterative recipe with early stopping based on VBench and human evaluation, which yields consistent gains over the baseline without inducing model collapse or severe overfitting to a particular reward. More discussions are presented in our response to Reviewer `gKUH`.


**7. All Vbench metrics**

Reviewer `gKUH` suggested to include all Vbench metrics for comprehensive comparison. Accordingly, we have inclucded the full set of VBench metrics for Wan-1.3B (baseline and Dual-IPO)in the appendix. The results consistently reflect the effectiveness of our proposed method.

**8. More human preference evaluation**

Reviewer `gKUH` noted that the human preferences are not shown for Wan2.1 experiments. Therefore, we present the human evaluation results on Wan2.1, both in the rebuttal and our revised version. The results demonstrate that our method (Dual-IPO) has clear advantages, with substantial gains in semantic consistency, motion smoothness, and Faithfulness.

Overall, we have included these details in our revised version and present more details in the rebuttal below to address all reviewers' concerns. We also plan to release all code, data, and models to ensure reproduction. We feel encouraged that Reviewer `pm1R` decided to increase the initial score from **4** to **6** after rebuttal and other Reviewers commented that most of their concerns have been addressed.
Thanks again for all reviewers' efforts and valuable insights that help us improve our paper.

Best,

Authors.

---

### Author Response · Authors · 2025-12-01
**Brief Summarization of Rebuttal and Revisions (Part1/2)**

We thank all reviewers for their valuable feedback, and their assessment of our work as "**comprehensive and well-motivated framework**" and "**strong data efficiency**" by reviewer `pm1R`,  "**conceptually elegant and empirically validated**" and "**methodological sound**" by reviewer `bijg`, "**the idea is interesting and novel**" and "**flexiblely supports both pairwise and pointwise supervision**" by reviewer `Qzdo`, "**attains a level of preference alignment that was not achievable with prior approaches**" and "**effective**" by reviewer `gKUH`. Regaring our experiments, our emprical results were acknowledged as "**extensive and captures the evolving nature of human preferences**" by reviewer `pm1R`, "**strong experimental validation shows consistent improvement**" by reviewer `bijg`, "**achieves new state-of-of-art performance, and the experiment validation is comprehensive and rigorous**" by reviewer `Qzdo`, "**even a small model is shown to surpass the performance of a larger one**" by reviewer `gKUH`.

We are grateful for the constructive suggestions given by all reviewers. Based on their suggestions, we have made several additions to the manuscript, which are highlighted in the revised version.
We briefly summarize the most significant rebuttal and additions below:

**1. Detailed description of textual prompt construction and bias mitigation**

Reviewer `pm1R` suggested including more details about the construction of textual prompts. To address this, we have added Figure 1 in the appendix to present the detailed distribution of the sub-categories for both subjects and actions.
To facilitate reproduction, we provide anonymous links to generate prompts as follows: https://anonymous.4open.science/r/ICLR2026_Dual_IPO-5262/create_prompt.py.
Besides, we added detailed description to clarify the **Size**, **Diversity**, **Balance** of our constructed prompts.


**2. Computational cost of our framework**

Both Reviewer `bijg` and Reviewer `Qzdo` raised the concern of computational cost. We clarify that the dominant practical bottleneck of our framework comes from video sampling speed and VBench evaluation time, rather than from the Dual-IPO algorithm itself. Additionally, Dual-IPO is explicitly designed to be small-lab friendly by reaching strong (often SOTA-level) performance with small generators and a lightweight, reusable critic, which is computationally efficient.


**3. Reward hacking or reward overfitting**

Reviewer `bijg` raised the concern of reward hacking. We do observe “reward hacking” in our experiments. Around the 3rd–4th SRPO iteration, the internal reward score kept increasing while VBench dropped, indicating that the generator was overfitting to the existing reward model. After updating the reward model with SRPO (using human-validated pseudo labels), this mismatch was significantly alleviated, and VBench recovered, showing that iteratively adapting the reward model is an effective way to mitigate such reward hacking. In later iterations (beyond ~6), we observe that VBench improvements become very marginal, even though the reward given by our critic continues to increase. We conjecture two reasons for such observation: (i) the current small T2V model may already be close to its performance ceiling under VBench, and (ii) the evaluation dimensions emphasized by our reward model may need to be adjusted, for example, by placing more weight on aesthetics and visual quality, where the VBench scores are relatively lower than other dimensions.

---

### Meta-Review · Area_Chair_6qkW · 2026-01-05

**Summary:**

This paper proposes a dual-loop method to improve text-to-video generation using iterative feedback. Reviewers praised the novel framework and the strong performance where small models beat larger ones. One main concern was potential bias in the text prompts used for training. Another issue was the high computing cost and lack of convergence analysis. The authors explained their prompt balancing strategy and provided cost trade-offs during the discussion. Most reviewers were satisfied with the new data and clarifications provided. Overall this is a solid paper and I encourage the authors to add the promised details in future.

**Reviewer Concerns:**

The rebuttal successfully addressed concerns about prompt diversity and missing evaluation metrics for the Wan model. However, the high computational cost remains a practical issue for smaller labs. A formal theoretical proof for the convergence of the iterative process is also still missing.

**Reviewer Scores:**

Reviewer pm1R raised their score to 6 after understanding the prompt bias mitigation. Reviewer bijg maintained a score of 6, accepting the trade-off between cost and quality. Reviewer Qzdo would likely maintain their score of 6, though they did not confirm the final convergence explanation. Reviewer gKUH kept their score of 6 after seeing the additional Wan model metrics.

---

### Decision · Program_Chairs · 2026-01-26

Accept (Poster)